# In vitro assembly, positioning and contraction of a division ring in minimal cells

**Shunshi Kohyama** [1,2], **Adrián Merino-Salomón**[1,2] **& Petra Schwille** [1] ✉

Constructing a minimal machinery for autonomous self-division of synthetic cells is a major goal of bottom-up synthetic biology. One paradigm has been the *E. coli* divisome, with the MinCDE protein system guiding assembly and positioning of a presumably contractile ring based on FtsZ and its membrane adaptor FtsA. Here, we demonstrate the full in vitro reconstitution of this machinery consisting of five proteins within lipid vesicles, allowing to observe the following sequence of events in real time: 1) Assembly of an isotropic filamentous FtsZ network, 2) its condensation into a ring-like structure, along with pole-to-pole mode selection of Min oscillations resulting in equatorial positioning, and 3) onset of ring constriction, deforming the vesicles from spherical shape. Besides demonstrating these essential features, we highlight the importance of decisive experimental factors, such as macromolecular crowding. Our results provide an exceptional showcase of the emergence of cell division in a minimal system, and may represent a step towards developing a synthetic cell.

Constructing a fully synthetic cell from defined biological molecules is one of the great aims of bottom-up synthetic biology[1–5]. Of the many essential features of cells, such as metabolism, replication, and interaction with the environment, cell division is probably the most tangible goal to be reached with a basic set of functional components, owing to the successful reconstitution and thorough mechanistic understanding of many division-related protein machineries[6–8]. One of the most advanced systems in this respect is the division machinery of the bacterium *Escherichia coli*[9,10]. In *E. coli* cells, three Min proteins called MinC, MinD, and MinE constitute a reaction-diffusion system that exhibits temporal oscillations between cell poles, so-called Min waves, generating a protein gradient that forms its maxima at the cell poles and minimum at mid-cell[11,12]. This protein gradient spatially regulates depolymerization of the division ring protein FtsZ, targeting proto-ring filaments to the middle of the cell by anchoring them to the membrane through FtsA and ZipA proteins, constructing a primary division ring known as FtsZ-ring[13,14] (Fig. 1a).

Remarkably, the self-organization of the Min gradient patterns and FtsZ polymerization-depolymerization dynamics have been reconstituted in vitro[15–26] on supported lipid bilayers (SLBs) and inside lipid compartments, such as microdroplets and vesicles,

thereby significantly contributing to a quantitative mechanistic understanding of these systems. FtsZ has been shown to polymerize into dynamic ring-like structures[17] that can deform free-standing membranes[18–22], and co-reconstitution of the Min positioning system and membrane-anchored FtsZ on supported membranes has confirmed the spatial regulation of FtsZ polymers by Min patterns[23–26]. However, the precondition for Min wave-guided assembly of an FtsZ-based ring-like structure with contractile ability is the functional co-reconstitution of the two systems in a closed and deformable membrane compartment such as lipid vesicles. This has so far not been accomplished due to the complexity of controlling the large number of components and environmental factors, as well as their intrinsic complex dynamics[10,14,27,28].

To master the multiple-protein system, several studies attempted the generation of Min waves and FtsZ structures via cell-free protein synthesis in vitro[19,20,26,29]. Although this approach avoids the complications imposed by protein purification, its challenge lies in the expression of multiple functional proteins at the right time and in the right concentration ratios[30,31] that were shown to be critical for the emergence of self-organization of the MinCDE system in vitro[29,32–34], and other divisome proteins in vivo[10,35,36]. Indeed, cell-free expression

[1]Department of Cellular and Molecular Biophysics, Max Planck Institute of Biochemistry, Am Klopferspitz 18, 82152 Martinsried, Germany. [2]These authors contributed equally: Shunshi Kohyama, Adrián Merino-Salomón. ✉e-mail: schwille@biochem.mpg.de

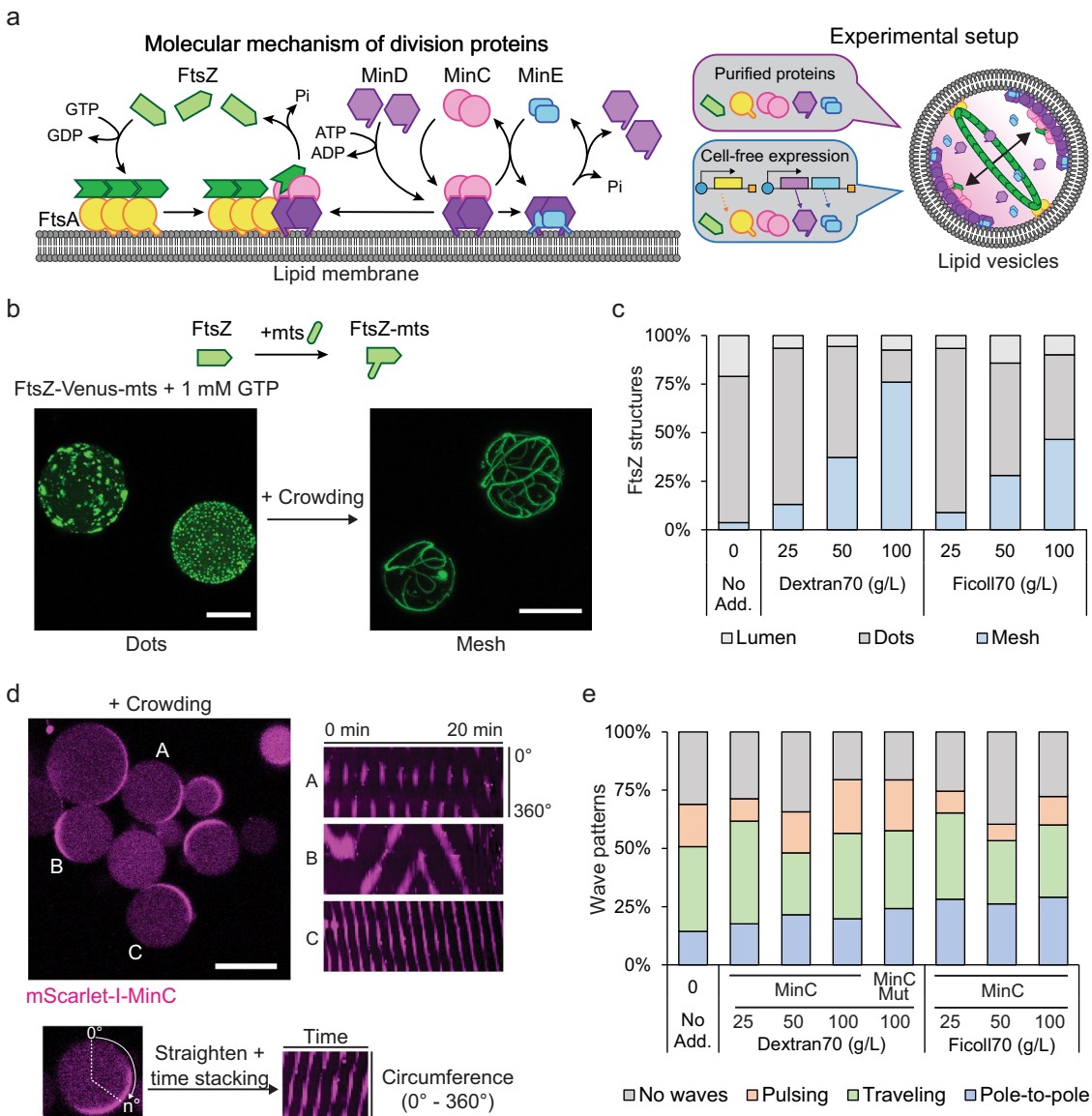

**Fig. 1 | Reconstitution of FtsZ or MinCDE systems inside lipid vesicles under macromolecular crowding conditions. a** Left: Schematic illustration of the molecular mechanism of the minimal bacterial division system. FtsZ polymerizes upon GTP hydrolysis and binds the membrane through FtsA to assemble into the FtsZ-ring. The dynamic MinCDE system self-organizes into oscillatory waves inhibiting FtsZ polymerization at cell poles. Right: Our experimental setup by which the minimal division system is reconstituted inside lipid vesicles using two different approaches: the purified components or the cell-free expression of the system. In both cases, we observe a minimal FtsZ-ring at the equatorial plane of the vesicle. **b** Top: schematic representation of FtsZ-Venus-mts. Bottom: 3D max projection of encapsulated 2 μM of FtsZ-Venus-mts (green) in the absence (left) or presence (right) of macromolecular crowding using 100 g/L Dextran70 and 1 mM GTP. Scale bars: 15 μm. **c** Frequency of FtsZ structures formed inside lipid vesicles. FtsZ structures were differentiated into three categories: Bundles assembling a mesh, FtsZ dots, or luminal localization ($n = 295, 230, 212, 242, 226, 218, 262$ for no additives, 25, 50, 100 g/L Dextran70, 25, 50, 100 g/L Ficoll70, respectively). We could not observe the assembly of FtsZ-rings regardless of crowding conditions. **d** Left: Representative

confocal image of purified MinCDE proteins inside lipid vesicles in the presence of macromolecular crowding (50 g/L Dextran70). Capital alphabet letters correspond to the kymographs on the right side (0.5 μM mScarlet-I-MinC, 3 μM MinD, and 3 μM MinE). Scale bar: 15 μm. Right: Kymographs of different MinCDE wave patterns captured by mScarlet-I-MinC fluorescence. The Kymograph is a 2D representation of a 1D object along with elapsed time. In this case, the fluorescence of mScarlet-I-MinC on the membrane is represented over time. The $Y$ axis represents the signal at the membrane of the circumference (however, circumference was straightened to present the 1D information) while the $X$ axis represents the time. Different intensity patterns observed in the Kymographs can be related with the Min oscillation modes on the membrane. **e** Frequencies of vesicles containing variations on the MinCDE dynamics (absence of waves, pole-to-pole oscillations, traveling waves, or pulsing) at different macromolecular crowding concentrations (0–100 g/L) using Dextran70 or Ficoll70 (0.5 μM mScarlet-I-MinC, 3 μM MinD, and 3 μM MinE) ($n = 374, 227, 181, 161, 297, 181, 298, 303$ for no additives, 25, 50, 100 g/L Dextran70, mScarlet-I-MinC^GIOD in 100 g/L Dextran70, 25, 50, 100 g/L Ficoll70, respectively). Images were collected after 10 min of vesicle preparation for 1–2 h.

of more than four division-related proteins within a single lipid vesicle has so far not been possible.

Herein, we successfully demonstrate Min wave-assisted FtsZ-ring assembly within lipid vesicles by two alternative approaches: in a fully controlled system with purified proteins, and by employing cell-free protein expression in a specifically tailored assay (Fig. 1a). In both

cases, we were able to follow by time-lapse imaging the origin, condensation, and equatorial placement of a minimal division ring-like FtsZ structure in giant unilamellar vesicles, more than ten times the size of bacterial cells. The more controlled reconstitution using purified proteins, MinCDE and membrane-anchored FtsZ allowed a quantitative assessment of decisive factors and revealed that crowding

environments are essential to form FtsZ-ring structures. As already suspected from our earlier work on MinDE wave-induced diffusiophoresis[25,37], we demonstrated that MinC, the main component for spatial regulation of FtsZ, was dispensable for assembly and spatial positioning of FtsZ polymers by MinDE, although the efficiency of ring formation was drastically improved by MinC. Moreover, we observed a positive feedback between the pole-to-pole oscillations of MinCDE and FtsZ-ring formation, in the way that the two processes promoted and spatially stabilized each other. On the other hand, our alternative approach using cell-free protein production within the vesicles supported a time-sensitive sequential expression of all the five components MinCDE, FtsA, and FtsZ. Strikingly, and in contrast to the system with purified proteins, this time-controlled series of events of FtsZ self-assembly and MinCDE oscillation resulted in a noticeable shape transformation of the spherical vesicle along with the Min-induced centric condensation of an originally isotropic FtsZ-FtsA meshwork. Our experiments thus reveal that such progressively condensed FtsZ-ring structures are not only able to constrict the vesicle precisely in the middle, but also induce a clear symmetry breaking of Min oscillations upon deviation from spherical geometry, selecting a single pole-to-pole oscillation mode that intensifies further equatorial FtsZ condensation in a positive feedback mechanism. These exciting findings emphasize how the timing of events may be crucial for unfolding a particular biological activity, and that under the right conditions, complex spatiotemporal biological dynamics can indeed be reconstituted in vitro, marking a significant step towards constructing synthetic cells from the bottom-up.

## Results

### Optimizing FtsZ and MinCDE reconstitution in lipid vesicles under macromolecular crowding conditions

Recent experiments with reconstituted proteins revealed that macromolecular crowding significantly influences the functionality of divisome proteins[32,38–40]. In particular, it has shown to promote lateral interactions among filaments, thereby enhancing FtsZ polymerization[39,40] and improving the regulation of FtsZ by Min waves into steeper gradients[23,41]. It has been suggested that the regulation of FtsZ by MinCDE, as well as FtsZ condensation into pronounced ring structures, is likely to depend critically on excluded volume effects[23]. Moreover, previous reports stated that FtsZ can only form bundles inside lipid vesicles under crowding conditions[18,22,42,43], which might be essential to achieve the assembly of a FtsZ-ring. On the other hand, much enhanced crowding with non-physiological molecules can easily become artificial and obstructive for the seamless interlocking of more complex biological machineries, in particular under cell-free conditions. Therefore, we first attempted an optimization of environmental conditions with respect to the functionality of the main components, MinCDE and FtsZ. To this end, we adapted an experimental setup and assay that stably generates Min waves via tuning MinE dynamics inside lipid compartments[29,32,33] and employed Dextran70 and Ficoll70, which are well-known and widely accepted macromolecular crowders in vitro[18,20,38,44,45]. For standardized vesicle production, we used a double-emulsion transfer method[29,46].

To simplify our experimental setup and avoid the use of membrane linkers of FtsZ, we employed FtsZ-Venus-mts, a commonly used FtsZ mutant containing the YFP variant fluorescent protein Venus and the membrane targeting sequence (mts) domain to confer the spontaneous membrane binding ability on FtsZ[16,18,21,22,47,48] (Fig. 1b and Supplementary Notes). First, we achieved the encapsulation of FtsZ-Venus-mts in lipid vesicles, showing a high proportion of FtsZ assembled into bundles, forming an isotropic mesh on the membrane in the presence of crowders (Fig. 1b, c, and Supplementary Fig. 1a). In contrast, the absence of crowders yielded only dot-like structures on the membrane (Fig. 1b, c, and Supplementary Fig. 1b), similar to previous observations[21]. The abundance of vesicles containing FtsZ bundles and

isotropic mesh structures increased considerably at 50 g/L or higher crowding concentrations (Fig. 1c and Supplementary Fig. 1a), which led us to select this concentration regime as optimal for efficient FtsZ bundling.

Next, we investigated the influence of different crowding conditions on Min wave dynamics inside lipid vesicles (Fig. 1d, e), which have been poorly addressed in the different experimental approaches so far[23,33]. To test this, we first verified the protein concentration dependence of Min wave dynamics in vitro, as it has been shown to be highly sensitive to MinD/MinE ratios on SLBs or in microdroplets[29,32–34]. Following established procedures, we successfully reconstituted Min waves using MinD, MinE, and mScarlet-I-MinC inside lipid vesicles (Supplementary Fig. 2a–d). In agreement with previous studies[32,33], we observed an overall higher prevalence of dynamic Min patterns, as compared to stationary localization in the lumen or at the membrane, when the MinDE ratios were roughly balanced (Supplementary Fig. 2b–d and Supplementary Notes). Then, we analyzed the impact of macromolecular crowding on the MinCDE patterns, successfully reproducing the above-described MinCDE dynamics inside lipid vesicles also at high crowding conditions optimal for FtsZ bundling (Fig. 1b, c, and Supplementary Movie 1), without significantly affecting the frequency of Min patterns (Fig. 1e and Supplementary Notes). Therefore, we could verify that both FtsZ and MinCDE systems are functional under macromolecular crowding inside lipid vesicles, a key precondition for their co-reconstitution towards the functional assembly of bacterial division machinery in vitro.

### Co-reconstitution of MinCDE and FtsZ under crowding conditions yields the assembly of pronounced FtsZ-ring structures

After characterizing the effect of macromolecular crowding over the FtsZ and MinCDE systems independently, we co-reconstituted both systems within vesicles at different crowding conditions (Fig. 2 and Supplementary Fig. 3). Intriguingly, for 50 g/L or higher concentrations of crowders, we observed an efficient condensation of previously isotropic FtsZ bundles into a single ring structure along with an equatorial positioning of this structure in the spherical vesicles, clearly driven by pole-to-pole oscillations of the MinCDE wave (Fig. 2b–e and Supplementary Fig. 3). These FtsZ-rings were slightly fluctuating spatially around the equatorial region of the vesicle as a consequence of Min oscillations. However, the ring structures remained stably positioned in the middle of the vesicles for an extended period of about half an hour, which constitutes a significant accomplishment towards assembling a functional divisome (Fig. 2c, d, and Supplementary Movie 2).

By analyzing the emergence of FtsZ structures inside lipid vesicles, we found that the formation of FtsZ-rings was highly related to the crowding concentration (Fig. 1c and 2f). The occurrence of FtsZ Mesh and FtsZ-rings increased with the concentration of crowders, and especially, it reached around 40% at 100 g/L Dextran70, while we did not find any FtsZ-ring structure in the absence of crowders or MinCDE proteins (Fig. 1c and 2f). In addition, the presence of FtsZ in the equatorial region of the spherical vesicles yielded a significant increase in pole-to-pole Min oscillations, indicating a tendency for mode selection by the presence of the FtsZ-ring structures or FtsZ bundles on the membrane (Fig. 2g). The establishment of Pole-to-pole oscillations under 100 g/L Dextran70 increased up to 43%, in good agreement with the yield of pronounced FtsZ-ring structures found under these conditions (40%) (Fig. 2f, g). Thus, we concluded that Min waves enhance and regulate FtsZ-ring assembly and placement with the help of macromolecular crowding in vitro.

### MinC is not required for FtsZ ring positioning

Interestingly, we observed that the presence of the FtsZ polymerization inhibitor MinC was not required for the positioning of the FtsZ-ring structure, as clear FtsZ-ring structures were also found in the

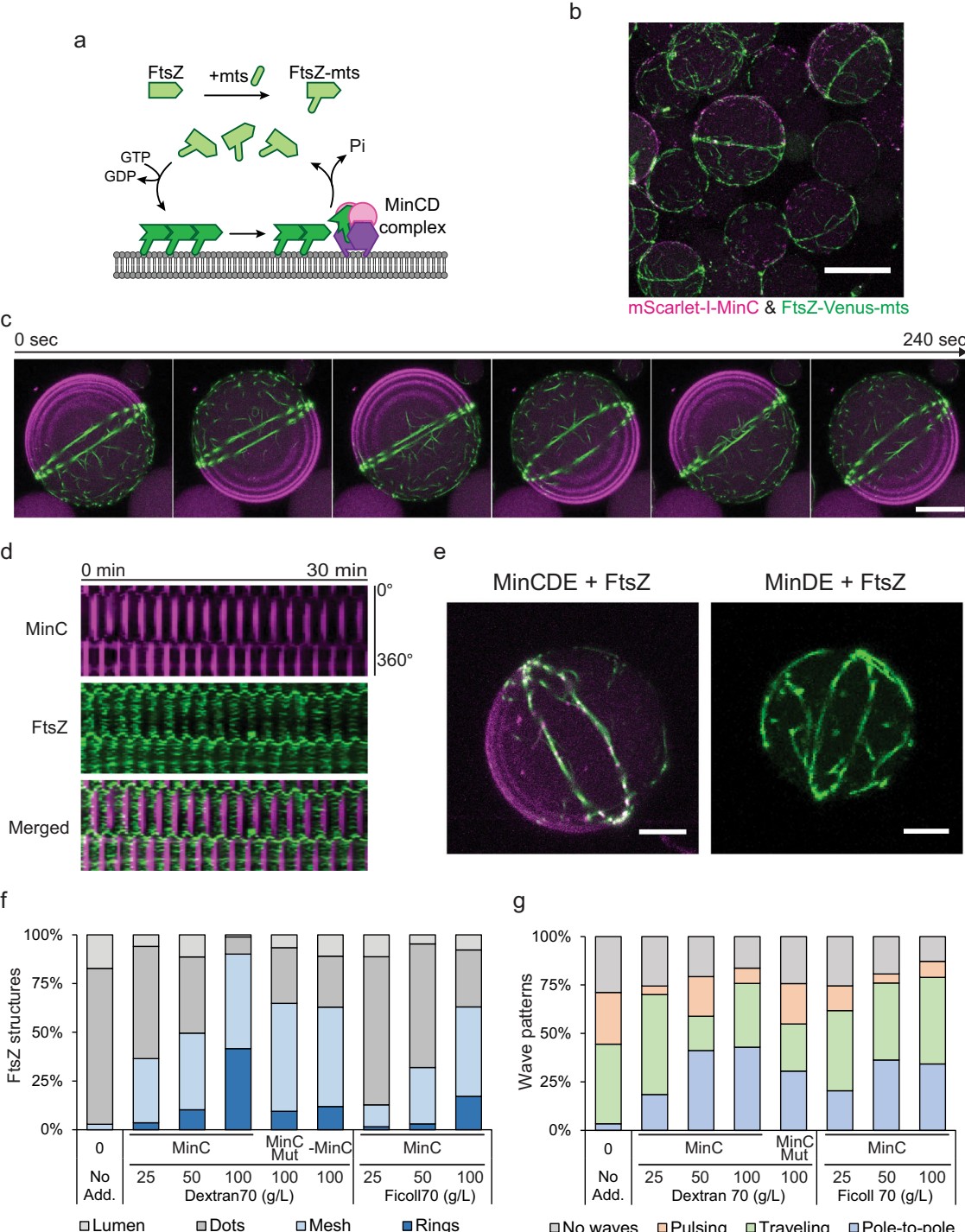

**Fig. 2 | FtsZ-ring assembly and positioning inside lipid vesicles. a** Schematic illustration of the FtsZ-Venus-mts system interacting with the membrane and the MinCDE system. **b** 3D max projection of a merged confocal image of vesicles containing the MinCDE proteins and FtsZ-Venus-mts in the presence of 100 g/L Dextran70. The Min wave-assisted formation of pronounced FtsZ-rings out of isotropic meshwork can be clearly observed. mScarlet-I-MinC and FtsZ-Venus-mts correspond to green and magenta, respectively (0.5 μM mScarlet-I-MinC, 3 μM MinD, 3 μM MinE, and 2 μM FtsZ-Venus-mts). Scale bar: 25 μm. **c** Time-lapse confocal images of the FtsZ-ring structure stabilized by pole-to-pole oscillations of Min waves. 3D max projection of merged images of mScarlet-I-MinC and FtsZ-Venus-mts. Scale bar: 15 μm. **d** Kymographs of the FtsZ-ring placement shown in Fig. 2c by a pole-to-pole oscillation wave captured by mScarlet-I-MinC (magenta) and FtsZ-Venus-mts (green). **e** (Left) 3D projection of an FtsZ-ring structure positioned by the MinCDE system. (Right) 3D projection of a vesicle containing a FtsZ-ring positioned

by the Min system in the absence of MinC. Min waves are not visible as mScarlet-I-MinC is absent in this case (3 μM MinD, 3 μM MinE, and 2 μM FtsZ-Venus-mts). Scale bars: 5 μm. **f** Frequency of FtsZ-ring formation in co-reconstituted vesicles. FtsZ structures were differentiated into four categories: rings, mesh, dots, or luminal localization ($n$ = 180, 252, 615, 473, 1045, 502, 250, 366, 543 for no additives, 25, 50, 100 g/L Dextran70, 100 g/L Dextran70 with MinC$^{G10D}$, 100 g/L Dextran70 without MinC, 25, 50, 100 g/L Ficoll70, respectively). **g** Frequency of Min wave patterns (pole-to-pole oscillations, traveling waves, pulsing, or absence of waves) inside vesicles containing the MinCDE system co-reconstituted with FtsZ-Venus-mts among the addition of different macromolecular crowding concentrations using Dextran70 or Ficoll70 ($n$ = 90, 157, 172, 289, 206, 227, 391, 289 for no additives, 25, 50, 100 g/L Dextran70, 100 g/L Dextran70 with MinC$^{G10D}$, 25, 50, 100 g/L Ficoll70, respectively). Confocal images were recorded after 10 min of the vesicle preparation for a time frame of 1–2 h. Source data are provided as a Source Data file.

absence of MinC at high crowding conditions (Fig. 2e, f). Moreover, similar results were observed using an FtsZ interaction defective MinC mutant (MinC[G10D])[41,49] (Fig. 2f). These observations agreed with previous studies that described a diffusiophoretic mechanism by which the Min proteins can position molecules on the membrane without any specific interaction[25,37]. Consequently, our results demonstrate that the diffusiophoretic positioning by the Min waves is preserved even in cell-like models on FtsZ. This calls for a greater future attention to the potential relevance of this mechanism in vivo, where its principle has been demonstrated recently[37].

However, it is important to point out that although a non-specific interaction between FtsZ and the Min waves could result in a distinct equatorial FtsZ-ring formation in vesicles, the presence of MinC significantly enhanced the efficiency of this process (Fig. 2e, f), showing FtsZ-ring formation with about four times higher abundance (From ~10–12% to ~40%), thus emphasizing MinC's role in the regulation of FtsZ polymerization. In contrast, MinC[G10D] did not increase the pole-to-pole oscillations as much as MinC in our experimental setup (from 25% to 31% for MinC[G10D], and from 20% to 43% for MinC) (Fig. 2g). This difference in the yield of pole-to-pole oscillations between MinC and MinC[G10D] suggests a positive feedback mechanism between Min oscillations and FtsZ positioning inside vesicles, in the way that the formation of the FtsZ-rings promoted by Min pole-to-pole oscillations in turn stabilize the pole-to-pole oscillation mode (Supplementary discussion).

### Increasing compositional complexity using PURE cell-free expression

Although the co-reconstitution of purified FtsZ-mts with MinCDE represents a step towards the assembly of a fully controlled minimal divisome, this system still lacks a potentially relevant degree of freedom in the regulation of FtsZ membrane attachment, which is in the cellular system conferred by anchoring through FtsA[17,50,51]. Membrane anchoring though another protein molecule rather than an mts motif on the FtsZ monomers not only decouples the stoichiometries of polymerization and membrane binding, but also confers more structural flexibility to the system. To investigate the potential relevance of this additional regulation feature, we set up the reconstitution of this more complex machinery inside lipid vesicles utilizing the PURE cell-free expression system[52,53]. Since transcription and translation are the basis of cellular information processing, combining gene expression with a rudimentary division system is an intriguing alternative concept for the bottom-up construction of synthetic cells. Thus, several studies have previously been performed to incorporate cell-free expression into a minimal division system in vitro[19,20,26,29]. Min waves were reconstituted inside lipid vesicles by de novo expression of MinDE proteins[26,29]. Also, FtsZ mesh structures and the deformation of lipid vesicles by FtsZ were observed by co-expression of FtsZ and FtsZ-related proteins, such as FtsA, ZipA, and ZapA[19,20] However, formation and placement of a distinctive division ring has not yet been accomplished by simultaneous expression of MinCDE, FtsZ, and FtsA. The expression of multiple genes from different, functionally interlocking self-organizing machineries is more challenging, due to the different temporal concentration ramps of expressed proteins.

To conduct cell-free expression within lipid vesicles, we used our standard protocol of vesicle formation and observation. Additionally, we employed a Peltier stage mediated temperature control device to incubate the vesicle-containing chambers for cell-free protein synthesis. After confirming the optimal crowding condition for our cell-free expression setup which allowed us to obtain about 10 μM of the reporter protein (superfolderGFP) expression (Supplementary Fig. 4 and Supplementary Notes), we first performed cell-free expression of FtsA with purified FtsZ-Alexa488 protein to visualize the integration of the wild-type FtsZ-FtsA cytoskeleton (Supplementary Fig. 5a). In accordance with previous reports[19,20],

under the Ficoll70 crowding condition FtsZ formed bundles on the membrane, anchored through cell-free expressed FtsA. Moreover, in addition to visualizing the readily formed FtsZ bundles at sufficient expression time (40–60 min), we also captured the process of FtsZ bundle formation on the membrane along with the expression of FtsA over 40 min (Fig. 3b, Supplementary Fig. 5b, c, and Supplementary Movie 3). The time-lapse images show the FtsZ dynamics inside lipid vesicles that gradually formed mesh structures on the membrane, with the bundles branching into even smaller ramifications (Supplementary Fig. 5b). Thus, we conclude that our experimental setup fully supports the transition to cell-free expression of our functional machineries, and that cell-free expressed FtsA enables us to capture the real-time dynamics of the development of an FtsZ-FtsA meshwork inside lipid vesicles.

### Direct observation of FtsZ-ring condensation inside vesicles

To test whether the FtsZ-FtsA cytoskeleton system also condenses into discernible FtsZ-ring structures inside vesicles upon spatial regulation by MinCDE waves, we next combined the expression of FtsA and MinDE (Fig. 3a). Since the *minDE* operon structure from the *E. coli* genome successfully resulted in Min patterns through PURE cell-free expression as previously reported[29], we constructed the *minDE* operon DNA template as well. Cell-free expression from this operon DNA template supported the MinDE expression system to produce Min patterns inside up to 80% of the vesicles (Supplementary Fig. 6c). We then attempted to express MinDE and FtsA with purified FtsZ-Alexa488 and mCherry-MinC to investigate the FtsZ dynamics under the regulation of Min waves.

Similar to the previous studies that indicate antagonistic membrane localization of FtsZ and MinCDE waves[23–26,41], the FtsZ structures dynamically reorganized in a time-dependent manner in response to the emergence of Min wave patterns (Fig. 3c and Supplementary Movie 4). At the beginning of protein expression, neither FtsZ nor Min protein patterns could be observed. Then, FtsZ gradually developed into mesh structures on the membrane, similar to the experiments without MinCDE. Only with a noticeable delay of more than 10 min after FtsZ mesh formation, Min waves emerged predominantly as pole-to-pole oscillations and instantly started to reorganize the FtsZ meshwork, eventually driving FtsZ to condense into a ring-like structure at the middle of the vesicle (Fig. 3c, d, and Supplementary Fig. 5f). Thus, we confirmed that partially cell-free expressed FtsZ-FtsA and MinCDE systems coordinate to assemble and correctly position FtsZ-ring structures inside lipid vesicles. However, the ring structure obtained through this process was not stable for an extended time period beyond 40 min. The Min oscillations eventually degenerated into traveling waves, and FtsZ mesh structures reappeared, however, FtsZ only antagonistically localized against the Min waves, resulting in a continuous spatio-temporal rearrangement of the ftsZ mesh (Fig. 3c and Supplementary Fig. 5f).

These time-dependent pattern changes of Min waves leading to a loss in mode-locked pole-to-pole oscillations were similar to the ones previously reported[32]. Therein, Min waves inside microdroplets showed pole-to-pole oscillation in the relatively early phase after Min wave emergence, but then deteriorated towards traveling waves due to the change in the MinDE concentration over time[29]. Indeed, some vesicles showed pole-to-pole oscillation only for less than 5 min and then transited to traveling waves (Supplementary Fig. 5f). FtsZ followed this trend and maintained pronounced ring structures only while pole-to-pole oscillations were present (Supplementary Fig. 5f). Together with the previous studies showing the reorganization of FtsZ by Min waves[23–26,41], it became obvious that Min waves strictly govern FtsZ patterns, and more importantly, out of the two major dynamic Min patterns, only pole-to-pole oscillations, but not traveling waves, support stable FtsZ-ring formation inside lipid vesicles.

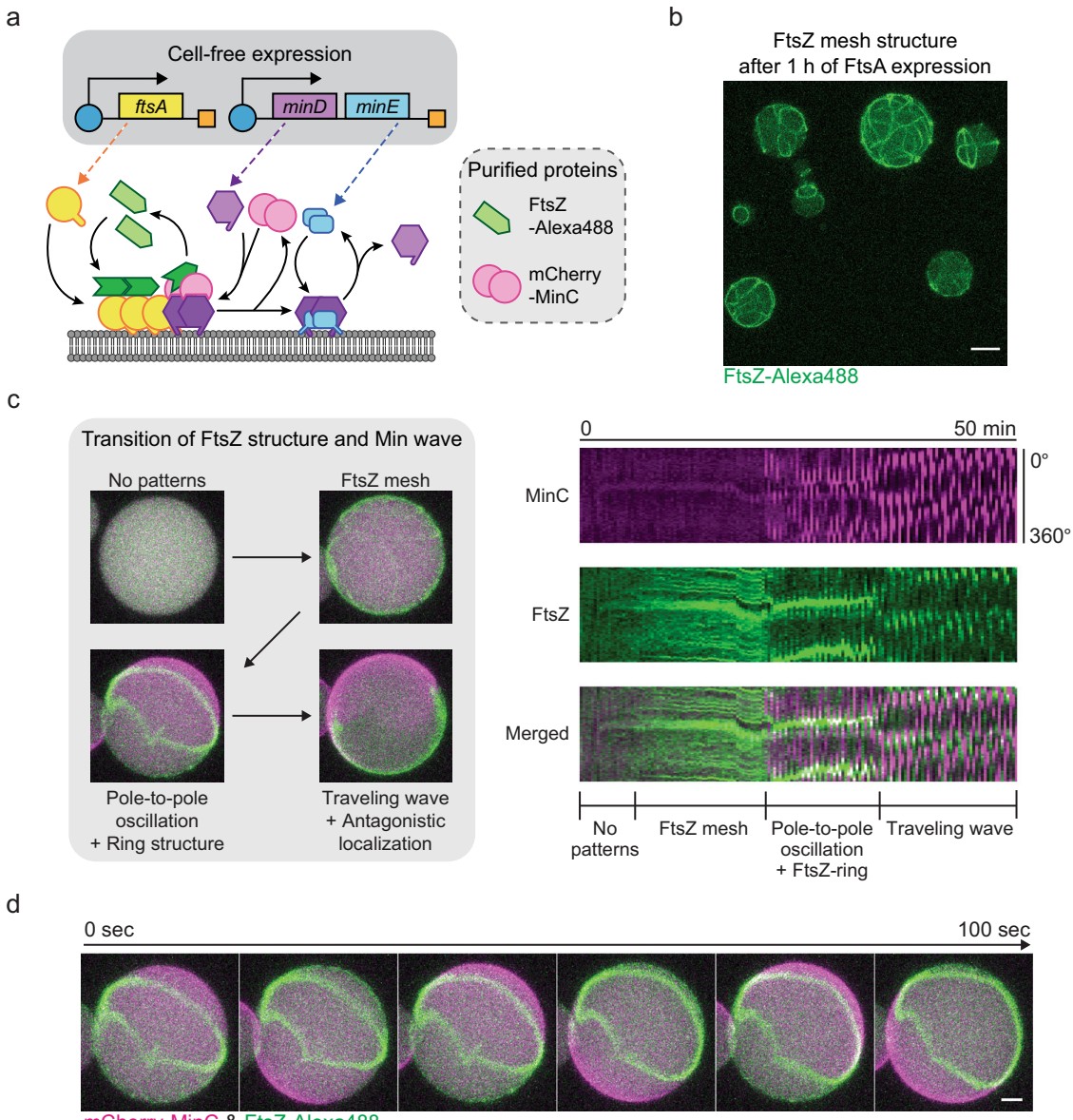

**Fig. 3 | Reconstitution of FtsZ-FtsA mesh and Min wave regulated FtsZ-ring structures via cell-free expression. a** Schematic illustration of PURE cell-free co-expression of FtsA, MinD, and MinE with purified FtsZ-Alexa488 and mCherry-MinC. **b** 3D max projection of FtsZ-Alexa488 mesh structures inside vesicles via FtsA expression (after 1 h of expression with 5 nM of *ftsA* template). Scale bar: 10 μm. **c** Regulation of FtsZ localization by Min waves. Left: 3D max projection of the transition of FtsZ structures and Min wave patterns by co-expression of FtsA, MinD, and MinE within vesicles (2 nM *ftsAopt* and 2 nM *minDEoperon* templates). Starting from no patterns, an FtsZ mesh develops on the membrane, which is then instantly condensed into an FtsZ-ring structure by the later onset of pole-to-pole oscillations of Min waves, through antagonistic localization of Min and FtsZ proteins. Right: Kymographs of the pattern transition of Min waves and FtsZ structures. Recording started after 5 min of cell-free expression. mCherry-MinC and FtsZ-Alexa488 are indicated in magenta and green, respectively. **d** Time-lapse images of FtsZ-ring like structures induced by pole-to-pole oscillations of Min waves. Experimental condition is the same as Fig. 3c and recording started after 30 min of cell-free expression. Scale bar: 5 μm.

## Ensuring compatibility of fully cell free expressed MinCDE and FtsZ-FtsA systems

After reconstitution of FtsZ-ring structures by partially cell-free expressed proteins, we expanded the system towards building the entire division machinery with genetically encoded proteins. One challenge for this approach is that either one or two proteins must be fused to Fluorescent Proteins (referred as FPs) to monitor their dynamics by fluorescence microscopy. However, *E. coli* FtsZ is known to lose its functionality by fusing tag proteins to the C-terminus[51,54]. Furthermore, an increase in the molecular weight of FtsZ by conjugating average-size FPs decreases the protein expression yield. On the other hand, since MinDE waves are highly sensitive to the protein concentration ratio, it needed to be validated that cell-free expressed MinCDE proteins can maintain ideal concentrations to emerge wave dynamics. Thus, we first reconstituted the partial systems (MinCDE vs. FtsZ-FtsA) and verified the self-organization ability of cell-free expressed proteins (Supplementary Fig. 6a).

Since the *minDE* operon DNA template had already been optimized for cell-free expression[29], we decided to keep the *minDE* operon structure for the MinDE expression. Therefore, we additionally introduced mCherry-MinC as reporter of the Min wave localization, because it would not be required for self-organization, and is at the same time the regulator for FtsZ localization. As expected, co-expressed MinCDE proteins sustained Min wave emergence inside

vesicles (Supplementary Fig. 6b and Supplementary Movie 5), implying that MinDE concentrations were still within a well-balanced range of co-expression (Supplementary Fig. 6b, c, and Supplementary Notes).

Next, we investigated the assembly of FtsZ-FtsA from fully cell-free synthesized FtsA and FtsZ, which promised to be more complicated than the MinCDE system. To overcome the loss of function in FtsZ-FPs chimeric proteins[51,54], we adapted a novel FtsZ mutant according to a previous report[55], in which FPs are inserted into the G55-G56 position of FtsZ instead of C-terminal conjugation. After the validation of FtsZ and FtsA template concentrations (3 nM for FtsZ and 1 nM for FtsA) to maintain the ideal protein proportions to form FtsZ bundles, we confirmed that co-expressed FtsZ-G55-Venus-Q56 and FtsA generated FtsZ mesh structures on the membrane (Supplementary Fig. 6d, e, and Supplementary Movie 6). Simultaneously, we captured the mesh formation of FtsZ-G55-Venus-Q56 that showed to be similar to the FtsA-only expression with purified FtsZ-Alexa488 (Fig. 3b and Supplementary Fig. 6d), demonstrating a similar correlation of FtsZ intensity on the membrane with vesicle size as observed for purified proteins (Supplementary Fig. 5e, 6h, i, and Supplementary Notes). Together with the reconstitution of the MinCDE system, we concluded that sub-division systems could be faithfully reconstituted from cell-free expressed proteins inside lipid vesicles.

### Mid-cell placement and radial constriction of a minimal division ring

Finally, we incorporated the co-expression of MinCDE, FtsZ and FtsA proteins under the established conditions within vesicles (Fig. 4a). We confirmed that upon co-expression of all five proteins, FtsZ-ring structures spatially self-assembled under the regulation of Min waves (Fig. 4b, c, and Supplementary Movie 7). As observed for the partial co-expression described above (Fig. 3c), FtsZ first assembled into isotropic mesh structures, which upon the onset of MinCDE oscillations spatially condensed into an FtsZ-ring structure in the middle of the vesicle (Fig. 4d, Supplementary Fig. 7a, c, d, Supplementary Movie 8, and 9). In turn, as observed for the purified proteins, the formation of the ring structure led to a mode selection of the Min oscillations into a pronounced pole-to-pole dynamics perpendicular to the plane defined by the FtsZ ring structure.

Strikingly, in contrast to all other assays with partly purified proteins, we found that the simultaneous evolution of FtsZ-ring assembly and mode selection into pole-to-pole Min oscillations resulted in a marked deformation of the lipid vesicles away from spherical to a more rod-like shape, reflecting on a radial constriction of up to 19% (Fig. 4f and Supplementary Fig. 8). This vesicle deformation by a central FtsZ-ring structure could also be observed for static bipolar localization of Min proteins but not for traveling waves, implying that the static pattern of Min waves also supports FtsZ-ring assembly and placement (Fig. 4c, Supplementary Fig. 7a, b, and Supplementary Notes).

Evaluation of the vesicle size dependence of FtsZ organization and vesicle deformation revealed more unexpected features of the reconstituted minimal division system. The population of deformed vesicles with pronounced FtsZ-ring structures increased with the vesicle size (22.5% at 12–16 μm, 40% at 16–20 μm, 50% at >20 μm), while relatively small vesicles (~12 μm) remained spherical (Fig. 4e). We also found that the aspect ratio of the deformed vesicles and their original diameters were moderately correlated (r-value = −0.40, Fig. 4f). In several cases such as shown in Fig. 4d, the aspect ratios (diameter/length) of the deformed vesicle approached about 0.75 after 3 h expression. Since we performed all experiments under isosmotic conditions, this suggests that the FtsZ-rings upon spatial condensation generated considerable forces to constrict the vesicles radially by up to 19% (Supplementary Fig. 8). Taken together, these results successfully demonstrate that such a complex biological function as contracting ring-based cell division could indeed originate from a set of well-defined components in vitro.

## Discussion

In this work, we have reconstituted the bacterial division ring placement system in vitro from a minimal set of components, namely, MinC, MinD, MinE, FtsA, and FtsZ by cell-free expression. At the right choice of template ratios, proteins have been orchestrated reliably within giant lipid vesicles to form FtsZ-ring structures, and unexpectedly, even shown to constrict their membrane containers, despite being more than an order of magnitude larger than bacterial cells. In addition to the complete cell-free expressed system, we have quantitatively investigated the assembly and spatial regulation of a FtsZ-ring with MinCDE and FtsZ-Venus-mts in a fully controlled system based on purified proteins, which has allowed us to evaluate the key factors involved in the process, such as optimal Min protein concentrations or macromolecular crowding conditions (Supplementary discussion). Both approaches highlight important features of the bacterial division mechanism in vitro, confirming the relevance of the coupled protein systems FtsZ-FtsA and MinCDE as paradigms for minimal cell division in bottom-up synthetic biology.

One of the expected insights gained from our study and in agreement with previous observations[41] is that MinCDE pole-to-pole oscillations indeed seem to be directly responsible for FtsZ-ring formation and placement in the sense that a previously isotropic mesh-like appearance of FtsZ filaments condensed into ring-like structures targeted to the equatorial region of the vesicles (Figs. 2c–e, 3d, and 4c, d). Not directly expected, but nevertheless highly plausible, was our observation that the condensation of FtsZ into ring-like structures, reminiscent of a spatial symmetry breaking, in turn led to a mode selection for oscillatory MinCDE dynamics in the sense that pole-to-pole oscillations perpendicular to the emerging FtsZ ring structures were favored over pulsing or travelling wave modes (Supplementary discussion). By using the FtsZ interaction defective mutant MinC^G10D or removing MinC from the system, we also found that the interaction between MinC and FtsZ is not necessary to form and position FtsZ-rings (Fig. 2e, f), although functional MinC significantly enhances the assembly of FtsZ-rings and the emergence of pole-to-pole oscillations. Moreover, Min self-organization dynamics and FtsZ-ring assembly and positioning appear to reinforce each other in a positive feedback mechanism towards division. Indeed, in most cases, FtsZ-ring structures and pole-to-pole oscillations persisted for a long time, supporting this hypothesis (Fig. 2c, d, and Supplementary Movie 2). This positive feedback resulting in spatial symmetry breaking and establishment of a defined axis along which division is particularly remarkable with respect to the originally spherical vesicles, quite distinct from the rod-like shape of E. coli bacterial cells. In previous attempts of targeting circumferential actomyosin ring structures with contractile features to spherical membrane vesicles[56], the lack of a defined division axis resulted in the slipping off from equatorial to polar regions upon ring constriction, in stark contrast of what can be observed in the presence of spatially stabilizing Min oscillations. We can thus conclude that any deviation from spherical symmetry leads to preferential mode selection of pole-to-pole oscillations.

The other highlight of our results was the pronounced vesicle deformation by the interplay of FtsZ-ring assembly and Min wave-induced positioning in the complete set of cell-free expressed proteins on the scale of tens of micrometers. Even though there were previous reports on how FtsZ and related proteins may constrict or deform lipid vesicles[16,18–22], our results allowed to visualize the gradual onset of force generation along with the condensation and positioning of a large FtsZ-ring from an isotropic meshwork in a plausible chain of causation (Fig. 2c, 3c, and 4d). Recent studies demonstrated that micron-sized treadmilling rings of FtsZ produce weak forces in the 1-2 pN range that can only constrict deflated

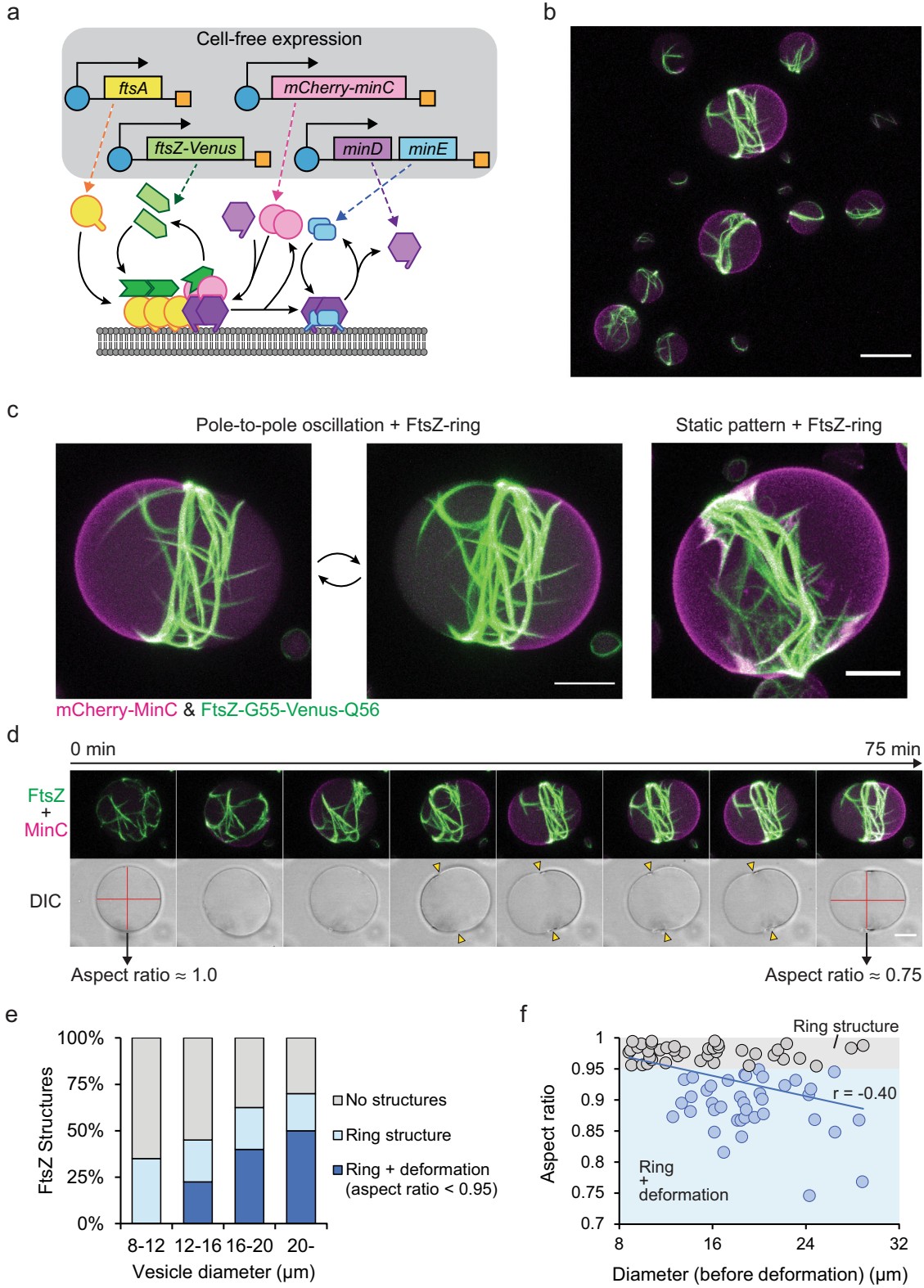

mCherry-MinC & FtsZ-G55-Venus-Q56

vesicles or wall-less *E. coli* cells[21], much lower than what is estimated for constriction in native cells (8-15 pN)[57,58]. For the equatorial FtsZ-rings that are more than an order of magnitude larger than any known FtsZ ring-like structure in vitro, we did not observe any deformation of GUVs under isosmotic conditions when FtsZ was directly attached to the membrane (Fig. 2c–e). In contrast, FtsZ-rings anchored to the membrane through FtsA constricted the vesicles (Fig. 4c, d). This suggests that the FtsZ-FtsA system might generate

greater effective forces on membranes, potentially owing to a different mechanism of dynamic membrane attachment, similar to the spatiotemporal imbalance of membrane curvature of lipid vesicles induced by attachment of MinDE proteins through their amphipathic helices[59,60], aiding large membrane deformations. In this regard, accumulation of transmembrane proteins that are also part of the bacterial divisome but have been missing in our system so far, such as ZipA, might enhance the local bending of the membrane as

**Fig. 4 | Reconstitution of minimal division ring placement system by co-expression of five essential components. a** Schematic illustration of co-expression systems of FtsA, FtsZ-G55-Venus-Q56, mCherry-MinC, MinD, and MinE. **b** 3D max projection of FtsZ-ring formation and Min protein localization at the pole of the vesicles after 100 min of co-expression at 37 °C (1 nM *ftsAopt*, 3 nM *ftsZopt-G55-Venus-Q56*, 2 nM *mCherry-MinC*, and 1 nM *minDEoperon* templates). mCherry-MinC and FtsZ-G55-Venus-Q56 are indicated in magenta and green, respectively. Scale bar: 20 µm. **c** Enlarged images of FtsZ-structure and Min wave patterns inside lipid vesicles after 3 h of cell-free expression. Pole-to-pole oscillations and static patterns stabilize the FtsZ-ring structures, and in both cases, FtsZ-ring can constrict vesicles. Experimental condition is the same as Fig. 4b. Scale bars: 10 µm. **d** Time-lapse images of the formation of the FtsZ-ring structure and constriction of the vesicle driven by the pole-to-pole oscillations of MinCDE. Top panels: 3D max projection of the merged image of mCherry-MinC (magenta) and FtsZ-G55-Venus-Q56 (green). The isotropic FtsZ mesh is condensed into a ring structure after the emergence of pole-to-pole oscillations, which later constricts the vesicle membrane. Bottom panels: Differential interference contrast (DIC) images of the

equatorial plane of the vesicle. Red lines indicate aspect ratios of the vesicle before and after constriction by FtsZ-ring, yellow triangles indicate necks of the constriction by FtsZ-ring. Experimental condition is the same as Fig. 4b and the recording started after 50 min of cell-free expression. Scale bar: 10 µm. **e** Percentage of the ring structure and deformed vesicles among different vesicle sizes after 2–3 h co-expression of 5 essential genes (at the same template ratios as Fig. 4b) at 37 °C. Proportions of both FtsZ-ring structures and deformation of the vesicles increase along vesicle diameter, while deformation never appears under 12 µm in a diameter. n = 40 for each diameter range. **f** Scatter plot of the aspect ratios of deformed vesicles against vesicle diameter. The correlation coefficient shows moderate correlation (Pearson's r = −0.40), suggesting greater effective force generation by FtsZ-rings within larger vesicles. Gray and blue dots (with respective background colors) indicate non-deformed vesicles (aspect ratio > 0.95) and deformed vesicles (aspect ratio < 0.95), respectively. n = 79 from three independent biological replicates. Experimental condition is the same as shown in Fig. 4e. Source data are provided as a Source Data file.

hypothesized previously[61], thus being interesting candidates to improve the reconstitution of bacterial cell division (Supplementary discussion).

In conclusion, although we successfully reconstituted the bacterial division ring placement system in vitro, and could even witness ring-induced shape transformation of the vesicles, further investigations are necessary to develop synthetic cell models that are truly able to autonomously self-divide in an energy-dissipating manner based on their intrinsic biological machinery. Despite the exact sequence of events in bacteria is not fully clear, there is a high plausibility for a major mechanical role of cell wall expansion in vivo, a feature that will be challenging to replace in minimal systems. Further, one of the most promising candidates for additional large-scale force-inducing elements to aid FtsZ-ring constriction is actually the nucleoid, which could exert strong effects on the membrane through its replication and segregation via biochemical, but also biophysical mechanisms, such as wetting and liquid phase separation[62,63]. Thus, moving from the reconstitution of a minimal division machinery, that has now been accomplished to a large extent, to a more holistic system which also comprises minimal genome replication, appears to be the next exciting goal of bottom-up synthetic biology.

## Methods

### Plasmid construction

Plasmids for protein purification and cell-free expression were constructed by seamless cloning or blunt end cloning method according to the provider's protocol. All enzymes for cloning were purchased from Thermo Fisher Scientific (Waltham, MA, USA). Briefly, DNA fragments were amplified with overlaps regions between adjacent fragments by PCR using Phusion High-Fidelity DNA Polymerase and origo primers (Sigma–Aldrich, St. Louis, MO, USA). Then, PCR products were treated with DpnI and combined using GeneArt Seamless Cloning and Assembly Enzyme Mix. For deletion of sequences, a DNA fragment amplified from the original plasmid was treated with DpnI, phosphorylated using T4 Phosphokinase, and then ligated with T4 DNA Ligase. All plasmids were propagated in *E. coli* OneShot TOP10 (Thermo Fisher Scientific) and purified from overnight culture using NucleoBond Xtra Midi kit (Macherey-Nagel GmbH, Duren, Germany). Constructed gene sequences were verified using Sanger Sequencing Service (Microsynth AG, Balgach, Switzerland). Detailed construction methods are described in Supplementary Methods, and all primers are listed in Supplementary Table 1.

### Protein purification

His-MinD[15], MinE-His[15], FtsA[17], FtsZ-Venus-mts[47] were purified as previously reported. Wild-type FtsZ was purified by calcium-induced precipitation[64] and covalently labeled in the amino groups of N-term

amino acid residue with Alexa Fluor 488 carboxylic acid succinimidyl ester dye (Thermo Fisher Scientific) as earlier stated[40]. superfolderGFP (sfGFP), msfGFP-MinC, mCherry-MinC, mScarlet-I-MinC, and mScarlet-I-MinC^G10D were purified as described previously[15,32]. In brief, *E. coli* BL21 (DE3) pLysS cells were transformed with pET28a-His-msfGFP-MinC, pET28a-His-mCherry-MinC, pET28a-His-mScarlet-I-MinC, or pIVEX2.3d-sfGFP-His and incubated in LB medium (with 50 µg/mL Kanamycin or 100 µg/mL Ampicillin) at 37 °C while shaking at 180 rpm. After an optical density at 600 nm reached 0.2-0.3, Isopropyl-β-D-thiogalactopyranoside was added at a final concentration of 1 mM to induce expression of the target protein. Cells were further cultured for 2–4 h and harvested. Consequently, cells were resuspended in Lysis buffer (50 mM Tris-HCl, pH 7.5, 300 mM NaCl, 10 mM Imidazole) and disrupted using a tip sonicator (Branson ultrasonics S-250D, Thermo Fisher Scientific). The crude cell lysate was separated by centrifugation for 30 min at 20,000 × *g* at 4 °C, and the supernatant was mixed with Ni-NTA agarose (QIAGEN, Hilden, Germany). The sample was incubated for 10 min at 4 °C while gently shaking and loaded into an empty column. Ni-NTA agarose was washed with Wash buffer (50 mM Tris-HCl, pH 7.5, 300 mM NaCl, 20 mM Imidazole, 10% Glycerol), and then the target protein was eluted with Elution buffer (50 mM Tris-HCl, pH 7.5, 300 mM NaCl, 250 mM Imidazole, 10% Glycerol). The buffer of the protein solution was exchanged with Storage buffer (50 mM Tris-HCl, pH 7.5, 150 mM GluK, 5 mM GluMg, 10% Glycerol) using Amicon Ultra-0.5 centrifugal filter unit 10 kDa (Merck KGaA, Darmstadt, Germany).

FtsZ-G55-Venus-Q56 was purified using a calcium-induced precipitation method similar as described previously[64]. *E. coli* BL21 (DE3) pLysS cells were transformed with pET11b-FtsZ-G55-Venus-Q56, and the cell culture was prepared using LB medium (with 100 µg/mL Ampicillin) as described above. After collecting cell culture, the pellet was resuspended in PEM buffer (50 mM PIPES-NaOH, pH 6.5, 5 mM MgCl$_2$, 1 mM EDTA) and disrupted using a tip sonicator. The crude cell lysate was separated by centrifugation for 30 min at 20,000 × *g* at 4 °C, and the supernatant was collected. The supernatant was mixed with 1 mM GTP and 20 mM CaCl$_2$ and incubated at 30 °C for 15 min to induce FtsZ bundles. Subsequently, the FtsZ bundles were pelleted by centrifugation for 15 min at 20,000 × *g* at 4 °C, and the supernatant was discarded. The pellet was again resuspended in PEM buffer, and the supernatant was collected after a centrifugation for 15 min at 20,000 × *g*, 4 °C. Then, precipitation and resuspension steps were repeated (total two precipitation-resuspension cycles) to purify further. The buffer of the protein solution was exchanged with Storage buffer using Amicon Ultra-0.5 centrifugal filter unit 50 kDa (Merck KGaA).

The concentration of the proteins was measured by Bradford Assay (Bio-Rad, Hercules, CA, USA), and protein solutions were

aliquoted, frozen in liquid nitrogen, and stored at −80 °C until further use.

## Preparation of crowding solution

Lyophilized BSA (Sigma–Aldrich, catalog number: A6003) was dissolved in Reaction buffer (50 mM Tris-HCl, pH7.5, 150 mM GluK, 5 mM GluMg) at approximately 100 g/L. Then, the buffer was exchanged with Reaction buffer to wash out residual molecules using Amicon Ultra-0.5 centrifugal filter unit 50 kDa. After washing steps, the concentration of BSA was measured by Bradford Assay. Ficoll70 and Dextran70 (Sigma–Aldrich) were dissolved in Reaction buffer, and their concentration was calculated from the weight of crowding agents and total volume of the solution (typically, the final concentration of the solution was 300–500 g/L). Crowding solutions were stored at −20 °C until further use.

## Preparation of lipid vesicles

1-palmitoyl-2-oleoyl-glycero-3-phosphocholine (POPC) and 1-palmitoyl-2-oleoyl-sn-glycero-3-phospho-(1′-rac-glycerol) (POPG) (Avanti Polar Lipids, Alabaster, AL, USA) were mixed at 7:3 molecular ratio dissolved in chloroform at 25 g/L. In case of visualizing the lipid membrane, 2.5 mg/L of ATTO655 labeled 1,2-dioleoyl-sn-glycero-3-phosphoethanolamine (DOPE) (ATTO-Tech GmbH, Siegen, Germany) was further mixed into the lipid mixture. 50 μL of the lipid mixture was dried by nitrogen gas stream. 10 μL of decane (TCI Deutschland GmbH, Eschborn, Germany) was added to the lipid film, and lipids were resuspended by vortexing shortly, and subsequently, 500 μL of mineral oil (Carl Roth GmbH, Karlsruhe, Germany) was added and vortexed for 1 min to prepare a lipid-oil mixture.

We used the double emulsion transfer method for vesicle production to yield giant unilamellar vesicles[29,46]. For reconstitution with purified proteins, both inner and outer solutions were prepared in Reaction buffer. For cell-free expression experiments, a homemade PURE solution I based on a previous report[53] was used as the outer solution. The osmolarity of inner and outer was measured using an osmometer (Fiske Micro-Osmometer model120, Fiske Associates, Norwood, MA, USA), and outer buffer was diluted to match the osmolarities between inner and outer solution.

Lipid vesicles containing purified proteins or cell-free expression systems were formed following the same methodology with slight differences. For the case of purified proteins, a 96-Well Flat-Bottom Microplate (SensoPlate, Greiner Bio-One GmbH, Kremsmuenster, Austria) was used for vesicle formation and visualization, while for the cell-free system, vesicles were formed using a 1.5 mL tube. For the encapsulation of purified proteins, emulsion solution was obtained from 2 μl of the inner solution and 100 μL of the lipid-oil mixture mixed in a 1.5 mL tube by tapping. 100 μL of the outer solution was added into a well in a 96 well-plate, and subsequently, 50 μL of the lipid-oil mixture was layered on top of the outer solution. Then, ~80 μL of the emulsion solution was further dripped on the multi-layered solution. For the encapsulation of the Cell-free system, emulsion solution was formed by 5 μl of the inner solution and 250 μL of the lipid-oil mixture in a 1.5 mL tube. 500 μL of the outer solution was added to a fresh 1.5 mL tube, and subsequently, 200 μL of the lipid-oil mixture was layered to form a lipid monolayer. 200 μL of the emulsion was added carefully to the multi-layered solution.

After multi-layering, lipid vesicles were obtained by centrifugation for 10 min at room temperature (for purified proteins) or 4 °C (for cell-free expression) at appropriate centrifugation force. Applied centrifugation force was varied depending on the density of the inner solution given by crowding agents. Typically, we used $3000 \times g$ for 10 g/L, $500 \times g$ for 60 g/L, and $300 \times g$ for 110 g/L of the density of the inner solution. After centrifugation, the oil phase and the supernatant of the water phase were discarded. The rest of the water phase (approximately 100 μL) was then gently mixed by pipetting to resuspend lipid vesicles, and 50 μL of the vesicle solution was restored in another fresh tube.

## Self-organization assays inside lipid vesicles with purified protein

For reconstitution of the MinCDE system, an inner solution containing different concentrations of MinD and MinE (0.25–3 μM) and 0.5 μM mScarlet-I-MinC, mScarlet-I-MinC$^{G10D}$ were mixed with 10 mg/ml BSA and 2.5 mM ATP in Reaction buffer. For the co-reconstitution of FtsZ and Min system, FtsZ-Venus-mts was added to this mixture at 2 μM in addition to 2 mM GTP to trigger its polymerization. In samples containing crowder, different concentration of either Ficoll70 or Dextran70 was added at 25–100 g/L as the final concentration. Assays using only FtsZ were carried out following the same methodology in the absence of the Min system and ATP. All proteins and crowders were previously dialyzed or diluted in Reaction buffer. Confocal images were collected after 10 min of vesicle preparation and they were used for up to 1–2 h per sample.

## Cell-free expression inside lipid vesicles

Cell-free expression was carried out using PURE*frex* 2.0 (GeneFrontier, Chiba, Japan) following the supplier's instruction. DNA templates were linearized from corresponding plasmids using PrimeSTAR Max DNA polymerase (Takara Bio, Shiga, Japan) with T7P-F and T7P-R primers (Supplementary Table 1). Linearized templates and original plasmids are listed in Supplementary Table 2. The inner solution consisted of PURE*frex* 2.0, linearized DNA templates, 10 g/L BSA, and 50 g/L Ficoll70. For expression with purified proteins, 2 μM FtsZ-Alexa488, 0.5 μM mCherry-MinC, or 0.5 μM msfGFP-MinC were additionally supplied to the inner solution. The concentration of DNA templates and ratios were optimized to obtain the Min waves patterns and FtsZ structures. All template conditions are listed in Supplementary Table 3. The crowding agent and its concentration were varied for the estimation of sfGFP expression yield.

A homemade chamber was prepared for incubation with the temperature control stage. Coverslips (Menzel Glasses, # 1.5, 22 mm × 22 mm, Thermo Fisher Scientific) were washed with 70% EtOH. Then, three imaging spacers (Grace Bio-Labs SecureSeal imaging spacers, 1 well, 9 mm × 0.12 mm, Sigma–Aldrich) were stacked on a glass slip. 20 μL of 10 g/ BSA solution was added to the chamber to passivate the glass surface and leave for 10 min at room temperature, and subsequently, BSA solution was discarded and then washed with outer buffer. 30 μL of vesicle solution was added to the homemade chamber, and it was enclosed by another coverslip and sealed. Then, the chamber was mounted to the PE120-XY Peltier system (Linkam Scientific Instruments, Surrey, United Kingdom) and placed on Zeiss LSM780 confocal laser scanning microscope (Carl Zeiss AG, Oberkochen, Germany). The temperature was set at 4 °C before observation and then kept at 37 °C for cell-free expression.

## Microscopy and image processing

Images of lipid vesicle samples were taken by a Zeiss LSM780 confocal laser scanning microscope using a Plan-Apochromat 20x/0.80 air objective or C-Apochromat 40x/1.20 water-immersion objective (Carl Zeiss AG). Fluorophores were excited using a 488 nm Argon laser (for Alexa488, sfGFP, msfGFP, and Venus), 561 nm diode-pumped solid-state laser (for mCherry and mScarlet-I), and 633 nm He−Ne laser (for ATTO655). Images were typically acquired with 512 × 512 pixel resolution, 10-15 Z-stacks with 1–2 μm intervals, and 15–20 s intervals for 10 min to 3 h. All recorded tiff images were processed, visualized, and analyzed using Fiji[65] (v1.53 f). Z-stacks were visualized in 3D max reconstituted images by the Z projection function. Kymographs were generated using a custom ImageJ macro script. In short, the vesicles

periphery was detected from manually drawn ROIs, and then straightened to obtain time-stacked straight-line images. Subsequently, lines were stacked into an orthogonal direction against the long axis of the images in order of elapsed time (Fig. 1d).

## Self-organization assay on SLBs

Preparation of the supported lipid bilayers (SLBs) was described previously[25]. Briefly, a plastic chamber was attached to a cleaned glass coverslip (Menzel Glasses) using ultraviolet-curable glue (Norland Optical Adhesive 63, Norland Products Inc., Jamesburg, NJ, USA). The slide was cleaned in an oxygen plasma cleaner (model Zepto, Diener electronic, Ebhausen, Germany) for 30 s at 50% power. Small unilamellar vesicles (SUVs) were prepared at a concentration of 4 mg/ml of 1,2-dioleoyl-sn-glycero-3-phosphocholine (DOPC) and 1,2-dioleoyl-sn-glycero-3-phospho-(1′-rac-glycerol) (DOPG) mixture (DOPC: DOPG = 7:3 molar ratio) in a buffer (25 mM Tris-HCl pH 7.5, 150 mM KCl, 5 mM $MgCl_2$) by sonication in a sonicator bath. To generate the SLBs, SUVs were added to the reaction chamber at a concentration of 0.5 mg/ml and incubated for 3 min on a 37 °C warm heating block. The SLBs were washed 10 times with a total of 2 ml in the same buffer in the absence of magnesium (50 mM Tris-HCl pH 7.5, 150 mM KCl) to remove excess vesicles.

Before self-organization assays, the buffer in the chamber was exchanged with Reaction buffer. Then, FtsA was added to the Reaction buffer at 0.25 μM. After ~2 min of incubation, a mixture of either FtsZ-wt and FtsZ-Alexa488 (30%) or FtsZ-G55-Venus-Q56 at 1 μM was added to the chamber. Subsequently, 1 mM ATP and 1 mM GTP were added and mixed carefully. SLBs were incubated for >10 min before visualization. Fluorescence imaging was carried out on an inverted custom-built TIRF microscope[66] with a UAPON 100x/1.49 oil-immersion objective (Olympus, Tokyo, Japan). Excitation of the sample was made using a 488 nm laser, and the fluorescence signal was detected on a CMOS camera (Zyla 4.2, Andor Technology, Belfast, Northern Ireland).

## GTPase assay of FtsZ

GTPase activity of FtsZ was measured by quantifying the inorganic phosphate with a colorimetric phosphate quantification assay (BIO-MOL GREEN kit, ENZO life sciences, Lörrach, Germany) for 140 s[48]. Purified FtsZ-wt or FtsZ-G55-Venus-Q56 were used at 3 μM in Reaction buffer, and polymerization was triggered by 1 mM GTP. 13 μL fractions were added to a 96-Well Flat-Bottom Microplate (UV-Star, Greiner Bio-One GmbH) every 20 s after addition of GTP and mixed with 37 μL of Reaction buffer and 100 μL of BIOMOL GREEN reagent, stopping the reaction. After ~10 min of incubation at RT, the absorbance at 620 nm was measured in a TECAN plate reader (Tecan Group Ltd., Mannedorf, Switzerland) at room temperature. Phosphate concentrations were calculated from a $Na_2HPO_4$ standard curve in Reaction buffer, and the GTPase activity reaction rate (V, mol Pi/mol FtsZ/min) was determined from the slope of the linear part of phosphate accumulation curves.

## Analysis of the wave patterns and FtsZ structures inside lipid vesicles

Wave patterns of MinCDE proteins inside vesicles were analyzed using a custom ImageJ macro script. Briefly, time-lapse images were used to detect the lipid vesicles and obtain a kymograph from the fluorescence intensity on the peripheral membrane. These Kymographs were classified and checked manually in order to calculate the appearance ratios of each wave mode at each condition. FtsZ structures were visualized and classified manually using the 3D max projection of the lipid vesicles containing FtsZ.

## Estimation of the sfGFP expression level within lipid vesicles

First, different concentration series of purified sfGFP (0, 0.1, 0.3, 1, 3, 10, or 30 μM) was encapsulated in vesicles, and the fluorescence intensity of sfGFP was measured (n = 30 individual vesicles) from confocal images for each concentration to obtain the standard curve (Supplementary Fig. 4a). Then, sfGFP was expressed inside lipid vesicles using 2 nM *sfGFP* template with different macromolecular crowding conditions (no additives, 50 g/L Ficoll70, 100 g/L Ficoll70, 50 g/L Dextran70, or 100 g/L Dextran70), and the fluorescence intensity of sfGFP was measured within 100 individual vesicles for 200 min with 1 min intervals. The box plots were obtained from fluorescence intensities at 200 min (for Supplementary Fig. 4g), and time-development of sfGFP expression level was calculated from fluorescence intensities at each time point (Supplementary Fig. 4b–f).

## Analysis of the FtsZ localization inside lipid vesicles along with FtsA expression

Cell-free expression of FtsA was performed with 2 μM FtsZ-Alexa488 (*ftsA* template was omitted for a negative control) for 60 min with 20 s intervals (total 180 time points, recording started after 20 min incubation at 37 °C). Fluorescence intensity of FtsZ-Alexa488 in the lumen or membrane at the equatorial plane was then measured with a representative vesicle and average intensities at each time point were plotted in Supplementary Fig. 5b or d. The same procedure was repeated to measure the intensity of FtrsZ-G55-Venus-Q56 on membrane in the co-expression experiment of FtsA/FtsZ-G55-Venus-Q56 and plotted in Supplementary Fig. 6h. In both cases, average fluorescence intensities of FtsZ on membrane among different vesicles were measured after cell-free expression (total 80 min) and plotted against vesicle diameter in Supplementary Figs. 5e and 6i.

## Estimation of the wave occurrence with cell-free expressed Min proteins

Cell-free co-expression of MinD/MinE or mCherry-MinC/MinD/MinE was performed with 0.5 μM purified msfGFP-MinC for 50 min with 20 s intervals (total 150 time points, recording started after 10 min incubation at 37 °C). Then, vesicles were randomly chosen (n = 116 and 100 for MinDE and MinCDE expression, respectively.) and the percentage of Min wave patterns inside lipid vesicles were calculated as (sum of the dynamic waves and static patterns)/ (vesicle number) at every 3 min time point and plotted in Supplementary Fig. 6c.

## Analysis of FtsZ structures and the aspect ratio of deformed vesicles

Before and after 2–3 h of co-expression of FtsA/FtsZ-G55-Venus-Q56/ mCherry-MinC/MinD/MinE, Z-stack images of Venus fluorescence were acquired with tile-scan function. Vesicles were classified into four size ranges (8–12, 12–16, 16–20, >20 μm in a diameter, n = 40 for each range) from the images before expression. The structures of FtsZ were then detected manually from 3D max projection images, and vesicles with FtsZ-ring structure were further analyzed to determine the aspect ratio. The aspect ratio of the deformed vesicles was calculated as (length of short axis)/(length of long axis). Also, the degree of deformation was calculated as (length of short axis)/(diameter of the vesicle before deformation). The status of the vesicles was classified into three (no ring structure, ring structure, ring structure + deformation (with < 0.95 aspect ratio)) and plotted in Fig. 4e. The relation of the aspect ratio of the vesicles with FtsZ-ring and the original vesicle sizes were further plotted in Fig. 4f, and the correlation coefficient was calculated among all vesicles plotted in the figure (n = 79).

## Surface plot of the FtsZ distribution inside a lipid vesicle

The kymograph of FtsZ distribution at an equatorial plane of the vesicle shown in Supplementary Fig. 7c was obtained using a custom

ImageJ macro. Then, fluorescence intensities of the FtsZ were measured at every 10 min from the kymograph. The intensity profiles were then plotted against relative position on the peripheral of the vesicle (0°–360°) in Supplementary Fig. 7c.

## Statistics and reproducibility

The high number of vesicles per sample and the protein behavior found in them allowed a deep characterization of the phenomenology described in our results in a reliable manner. Wide field images showing multiple vesicles and several micrographs of similar results are shown in the figures to demonstrate and support the high reproducibility of our data together with quantitative analysis. In addition, all the micrographs shown in figures both in the main manuscript and supplementary file correspond to a reproducible result from 3 or more independent biological replicates.

## Reporting summary

Further information on research design is available in the Nature Research Reporting Summary linked to this article.

## Data availability

The data sets for all experimental conditions, graphs, and oligonucleotide sequences generated in this study are provided in the Supplementary Information/Source Data file. The original image data are available under restricted access for their large file size (>10 GB), access can be obtained by reasonable request from the corresponding author. Source data are provided with this paper.

## Code availability

The custom ImageJ macro code for wave pattern analysis and generation of kymographs has been deposited in github [https://github.com/ShunshiKohyama/Min-wave-analysis-in-GUVs].

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

## Acknowledgements

Special thanks to Germán Rivas (CIB Margarita Salas, CSIC, Spain) for providing essential scientific advice on the self-organization with purified proteins under crowding, and his helpful comments and critical reviews on the manuscript. We are grateful to Lei Kai (School of Life Sciences, Jiangsu Normal University, China) for the plasmid pIVEX2.3d-sfGFP-His, Leon Harrington and Tamara Heermann for their help in plasmid construction. We would like to thank MPIB Core Facility for assistance in protein purification, Michaela Schaper for plasmid cloning, Katharina Nakel for GTPase assay, Sigrid Bauer for lipid preparation, Kerstin Andersson for protein purification, and Michele Russo for assistance with the purification of FtsZ-G55-Venus-Q56. We are also grateful to Jan-Hagen Krohn and Yusuf Qutbuddin for assistance in TIRF microscopy. Further, we thank Leon Babl, Hiromune Eto, Ana Yagüe Relimpio, and Henri Franquelim for helpful discussions. This work has been supported by JSPS Overseas Research Fellowships (S.K.), the Max Planck-Bristol Centre for Minimal Biology (A.M.-S.), and CRC 863 (Forces in Biological Systems) by the Deutsche Forschungsgemeinschaft (P. S.). A.M.-S. is part of IMPRS-LS.

## Author contributions

S.K., A.M.-S., and P.S. conceived the study and wrote the manuscript. S.K. designed, performed, and analyzed all experiments with cell-free expression. S.K. made the custom ImageJ macro script for the analysis and kymograph visualization. A.M.-S. designed, performed, and analyzed all experiments with purified proteins within vesicles and SLBs. S.K. designed plasmids for cell-free expression and purification of sfGFP, msfGFP-MinC, mScarlet-I-MinC, mScarlet-I-MinC$^{G10D}$, mCherry-MinC, FtsZ-Venus, and FtsZ-G55-Venus-Q56. S.K. also purified sfGFP, msfGFP-MinC, mScarlet-I-MinC, mCherry-MinC, and FtsZ-G55-Venus-Q56. A.M.-S. purified and labeled FtsZ-wt. A.M.-S. also characterized FtsZ-G55-Venus-Q56 and FtsZ-wt by performing a GTPase assay.

## Funding

## Competing interests

The authors declare no competing interests.
