## [Peer Review File · Nature Communications]

In vitro assembly, positioning and contraction of a division ring in minimal cellsREVIEWER COMMENTS

Reviewer #1 (Remarks to the Author):

In this technically advanced study, the authors reconstitute parts of the bacterial cell division machinery in liposomes. They combine the MinD/C/E system with FtsZ/A using either purified proteins or in vitro expressed proteins. They observe various dynamical spatiotemporal organizations of these proteins on the membrane, among them central FtsZ ring formation and MinC oscillations. They empirically optimize conditions, including exploring effects of crowding agents, with the aim to observe more robust central FtsZ ring formation in the liposomes. An interesting result is that liposomes deform as central FtsZ rings seem to start contracting when the complete system is put together, going beyond previously observed less spatially controlled liposome deformations. This work represents a technical advance over previous similar reconstitutions that were either simpler in complexity or did not show MinCDE-dependent central FtsZ ring formation as convincingly as here. Surprisingly, the authors do not explain very much in mechanistic terms why the optimal conditions they find are optimal, although these systems have been studied quite extensively (separately) in the past and are at least to a certain extent well understood. Therefore, it appears that the focus of the work is not to gain new mechanistic insight into the bacterial cell division apparatus, but rather to push the limits of what's currently technically possible with respect to reconstituting a challenging in vitro system displaying interesting dynamical behavior, in part similar to some aspects of bacterial cell division.

Major concerns/questions/suggestions:

1. It is often not clear which concentrations of purified proteins are used in the experiments and which conditions are used for the in vitro expressions. To allow reproducing these experiments, the critical information defining the specific conditions shown (e.g. protein concentrations) should be clearly stated in the legends to complement the otherwise detailed information provided in the Methods.
2. Fig. 1c: Is this phase space expected given the current knowledge of the MinC/D/E system? If yes, please explain why (including references). If no, can a mechanism be proposed?
3. Suppl. Fig. 2c: Are ever FtsZ rings observed in the absence of MinC/D/E? If not, that should be stated for clarity. If yes, this should be included in the quantification.
4. Fig. 2c: FtsZ rings form in the presence of MinC/D/E. Has it really been demonstrated that FtsZ ring formation necessarily requires MinC oscillations? For example, can it be excluded that simply stronger bundling of FtsZ filaments due to the simple presence of MinC/D/E leads to ring formation? Similarly: is

centering of the FtsZ ring really due to the oscillations or does the ring simply form where the diameter is largest to minimise the bending energy of the filaments? The authors probably have the answers and could state them explicitly.

5. Similar question concerning the increased frequency of MinC oscillations in the presence of FtsZ: could it simply be the presence of FtsZ instead of an FtsZ ring that promotes MinC oscillations?

6. The authors suggest that there are mutual positive feedbacks between FtsZ ring formation and MinC/D/E oscillations. Is this expected based on the current understanding of these systems from previous work? If yes, that should be explained. If not, can the authors propose a mechanism?

7. Do the authors know how protein concentrations for the various proteins develop over time in the in vitro expression experiments? Do the observations in these experiments agree with those in the experiments with purified proteins, given the development of the protein concentrations over time?

8. Fig. 3: FtsZ ring formation is less stable here, as the authors observe, which may be the consequence of changing protein concentrations over time. Based on the current understanding of these dynamical systems, can a mechanistic explanation be proposed?

9. Fig. 5: Why are in these experiments multiple FtsZ rings observed, whereas in experiments without FtsA single rings could be seen?

10. Why are larger liposomes more strongly deformed by FtsZ rings than smaller liposomes. Which biophysical parameter could influence this deformability?

Minor comments:

1. Line 48: "low surface-to-volume ratio" - compared to what? Some people might think that the ratio is high, depending on which comparison they have in mind.

2. Line 101: It is not clearly stated from the start that purified proteins are used for the first experiments.

3. For all figures: Often it is not stated at which times after start of assembly are the displayed microscopy images are taken, which appears however important as the studied systems develop over time.
4. Fig. 1b. To make it easy, it could be stated in the figure which protein is shown. For the non-expert, it should be explained how the kymographs are generated: 1D lines are shown over time, generated from a 3D object. The presented microscopy images are rather small.
5. Fig. 1c. Example data for the 3 categories could be shown, at least in the Supplement.
6. Citations. Line 166. Reference 50 is a bit odd, because a review is called to support a rather detailed technical statement. Was this intended? Same for other reviews that are cited: can original literature be cited more specifically?
7. Fig. 2e. Please show some representative original image data for the different categories.
8. Line 240: Instead of only stating that "division ring placement" has not yet been achieved, it would be useful to also say what has been achieved so that the advance here is clear.
9. Line 420: Can the evidence for "constriction" (in 2 dimensions) instead of "squashing" (deformation in only 1 dimension) be presented?
10. Line 428: what does "some sort of dissipative mechanism" mean? A more precise statement could be clearer.
11. Line 440: "radically" - really? A more objective way to present the result would be to just state the quantity of the change.
12. Line 485: why is it so "unexpected" that FtsZ can cause constriction? Based on what? It is implied that constriction is rather expected for smaller liposomes. Based on which mechanistic argument?
13. Some jargon/abbreviations could be explained/introduced for the non-expert: e.g. SLBs, sfGFP,

Reviewer #2 (Remarks to the Author):

The manuscript titled “In vitro assembly, positioning and contraction of a division ring in minimal cells” by Kohyama and co-authors reports the reconstitution of a cellular division mechanism comprised of FtsZ/FtsA and MinEDC cell division proteins either purified or cell-free synthesized and encapsulated inside liposomes of sizes comparable to living cells. The MinEDG system exhibits oscillations inside the liposomes and a slight contraction of the membrane of liposomes is observed due to the assembly of FtsZ/FtsA rings. Min-wave assisted FtsZ-ring assembly within liposomes is observed by two different approaches, (1) one with purified proteins and (2) one using a cell-free expression system, the PURE system. The importance of macromolecular crowding in solution during the process of FtsZ ring formation is observed and discussed in the case of purified proteins. The occurrence of FtsZ rings increases with the concentration of crowders (here Ficoll and Dextran) with purified proteins. When proteins are synthesized using the PURE cell-free system, similar results are obtained.

Reconstituting molecular mechanisms in cell-free conditions enables to separate their biochemical and biophysical properties from the complexity of real living cells. This approach has become quite popular in bioengineering and constructive biology. Reconstituting a cell division mechanism, such as the one found in bacteria, is especially important as it is a critical step towards building artificial cell systems, an even greater undertaking for which major research programs have been launched on different continents. The experiments are in general well done and well described. That said, the manuscript has several major issues, the two major ones being that (1) this work is very close to an article published recently and consequently it does not seem to bring substantial new information, and (2) the manuscript is often confusing as it lacks a clear message about its novelty.

Major concerns:

- the work presented here is very close to the studies published recently by Godino et al 2019. It is difficult to understand where the novelty is. Phenomenologically, there is nothing really new compared to what Godino et al 2019 have published.
- the manuscript is often confusing, as there is no clear path in the setup of the experiments. The number of experiments presented is too large, which dilutes the message. Trying to publish a scientific study does not consist of flooding a manuscript with a load of experiments hoping that it looks good and impressive. Yes, a lot of work is presented in this manuscript, but it lacks some coherence. The manuscript should be shortened, the experiments condensed to gain clarity, and the novelty of the work should be clearly explained.
- molecular crowding in solution is optimized and discussed only for the system based on purified proteins and not with the PURE cell-free expression system. It is known, for instance, that crowding agents can impact the processes of transcription and translation as well as the self-assembly of cytoskeleton protein. Therefore, the optimum crowding conditions could be completely different from the one observed with purified proteins. Macromolecular crowding optimization should be done with the PURE system (see other concerns), or at a minimum discussed properly when the PURE system is used.

- it is never clearly explained why macromolecular crowding is studied in solution and how it could affect proteins that are predominantly located/bound at the membrane.
- what is the point of presenting two approaches, one based on purified proteins and one on cell-free expression? This renders the message of the work confusing. The cell-free protein synthesis approach seems more relevant in the context of artificial cells.
- It is also not clear how/why the five proteins were chosen. They do belong to the bacterial division mechanism. But many other proteins are involved in the Fts-ring formation process.

Minor concerns:

- with purified proteins, macromolecular crowding appears to be essential for Fts-ring formation. The crowders used are Dextran and Ficoll, both routinely employed in vitro to emulate crowding in biological solutions. PEG, the most popular and characterized crowding agent, is not mentioned. This should be discussed.
- considering that the proteins used or expressed are located at the membrane, it would have been interesting to also study the impact of molecular crowding at the membrane in the Fts-ring formation.
- Fig 3c: is it possible to estimate the concentration of proteins produced?
- a table that summarizes the frequency of the different patterns (rings, dot ...) at different concentrations of crowding agents with purified or expressed proteins would be useful.
- the discussion section should be shortened. It is too long and not very clear. The discussion should include just a few clear points and take-away messages.
- it would have been interesting to determine if the lipidic composition affects the assembly and the positioning of the ring within the liposomes. This should be discussed.

Reviewer #3 (Remarks to the Author):

This manuscript reports the complete reconstitution of the *E. coli* FtsA-FtsZ polymer system and the MinCDE spatial regulatory system in giant unilamellar vesicles, starting with Min oscillations that corral FtsZ-FtsA polymers into a medial zone, which ultimately leads to a small degree of membrane constriction at mid-liposome that may mimic the constriction forces at midcell in *E. coli*. The main improvements over previously reported bacterial divisome reconstitutions are the ability to synthesize all 5 proteins in a cell free system from DNA templates, following the transformation of a randomly localized FtsZ-FtsA polymer bundle within the liposome into a narrow band at the medial position over time, and the ability of this band to partially constrict the liposome membrane. Moreover, the data

suggest that not only do Min oscillations spatially constrain FtsZ-FtsA to a medial band, but also the FtsZ-FtsA medial band enhances the stability of pole-to-pole Min oscillations, which can also devolve into other types of movements that are generally not observed in vivo. The frequency of liposomes exhibiting these behaviors seems to be higher in this study than in previous studies, and the effect of specific crowding agents has been optimized. Together, these findings constitute several useful technical and conceptual advances for our understanding of how bacterial cell division can be reconstituted from minimal components.

Despite these strengths, there are several rather significant weaknesses. These include a lack of proper explanations in some cases, and too much emphasis on data that largely confirm previous studies instead of breaking new ground. For example, Zieske and Schwille (2014) already showed that oscillating MinCDE could focus FtsZ polymers into a fairly tight medial band in cell-shaped lipid microcompartments, but oddly that report was not cited here. As a result of these confirmatory experiments, many of the figures seem quite repetitive.

Perhaps most importantly, the large continuous FtsA-attached FtsZ polymer bundles that completely encircle the liposomes here are very different from the tight complexes of treadmilling FtsZ-FtsA observed at the septum of dividing *E. coli* cells. Consequently, it is not clear whether the membrane deformation observed here is in any way relevant to forces on the membrane in walled bacterial cells such as *E. coli*.

Hopefully the comments below will be helpful to improve the impact of the paper.

Major comments:

- 1) The authors do not adequately explain the different types of Min oscillations to the potential general readers of Nat Comm. In particular they need to explain what “pulsing” is.
- 2) Some of Fig. 1 and the first part of the Results are confirmatory—e.g. the need for the MinD:E ratio to be 1:1, the static membrane localization if ratio is too high, or lumen localization if the ratio is too low. Although these confirmatory experiments show that the system is working as expected, they do not advance our understanding of the phenomena being studied.
- 3) Lines 229-231: this is not a major advance, as co-reconstitution of purified FtsZ-mts with MinCDE was done by Zieske and Schwille (2014).
- 4) Line 232: In vitro reconstitution with FtsA and FtsZ was done in ref. 23.
- 5) Line 265-267: It seems that Ref. 31 already reconstituted the “cytoskeleton system” inside lipid vesicles via cell-free expression. If not, the authors need to explain how their advance is significant.

- 6) Line 284: the causality at this point is not fully backed by evidence, although later (line 297, supplementary Fig. 4d) there is a good correlation between pole to pole oscillations and proper FtsZ localization.
- 7) Line 288: are the traveling waves described here equivalent to the “circling” patterns made by Min proteins in liposomes described in Ref. 31?
- 8) Line 296-300: none of these findings/conclusions are surprising or new.
- 9) FtsA seems to be required for the membrane deformations in larger vesicles, as FtsZ-MTS on its own did not deform them. However, the dynamic treadmilling of FtsZ within the polymer bundles in the vesicles probably drive membrane deformations.
- 10) Do the authors think that the FtsZ swirls observed when bound to FtsA in SLBs are also occurring on liposome membranes? Given the larger scale of the polymer bundles compared with the swirls, this seems unlikely, but then how relevant are the swirls or the straight polymer bundles to what happens in vivo?
- 11) Line 426 and following: do the authors have an explanation for why pole to pole oscillations transition into traveling waves and then static localization? Does MinE specifically lose function or get degraded over time? Does the ATP in the system get exhausted?
- 12) It is not clear how Min oscillations initiate in perfectly spherical liposomes with an aspect ratio of 1.0, as there is no defined long axis until there is some deformation as shown by the authors, and compartment geometry has been shown previously to determine the orientation and organization of Min oscillations. How do back and forth oscillations get started without some asymmetry?
- 13) Line 493-494: What is an “expected insight”? MinCDE pole-to-pole oscillations are already known to be directly responsible for FtsZ ring formation and placement in vivo (and by Zieske and Schwille 2014, among others).
- 14) Line 501-503: Does the positive feedback on Min oscillation caused by the condensed FtsZ band depend on MinC? The prediction would be that it would be MinC-dependent given the direct interaction between MinC and FtsZ.
- 15) As discussed on lines 533-534, the membrane deformation with FtsZ-FtsA inside liposomes reported here is basically the same as the membrane deformation with FtsZ-FtsA* rings in liposomes reported in ref. 23, but just more frequent and efficient. Even the tubular liposomes reported 14 years ago in ref. 17 with FtsZ-MTS-Venus exhibited focused FtsZ “rings” that sometimes deformed the membranes to create a constriction.
- 16) Line 548: The decreased surface-to-volume ratio in larger spheres is an attractive hypothesis to explain the greater tendency of larger liposomes to undergo FtsZ-mediated constriction. However, it remains puzzling how extensive FtsZ bundles on the liposomes of very large diameters 10x the size of an E. coli cell could exert constriction forces, yes extensive FtsZ bundles of liposomes that are only modestly smaller (but still much larger than bacteria) would not.
- 17) Lines 559-560: “spatiotemporal imbalance of membrane curvature” needs to be explained better.

18) Line 562-564: The potential of ZipA to enhance membrane deformations by FtsZ/FtsA is not explained in sufficient detail, particularly as E. coli can still divide quite normally in the absence of ZipA when suppressor mutant divisome proteins are made.

19) The supplemental movies are very interesting to watch, and it is impressive how many liposomes have successful FtsZ medial localization. However, in the final two movies, it is puzzling why the largest and most prominent liposomes, initially still, start getting jostled around when the Min oscillations start (or perhaps vice versa). Can the authors explain this sudden onset of jerky liposome motions that coincide with the beginning of Min oscillations (and the focusing of FtsZ polymers towards the midpoint)?

Minor comments:

- 1) L. 243 should be “device to”
- 2) L. 256 should be “...process of FtsZ bundle...”
- 3) L. 261 should be “...the vesicle was gradually decreased...”
- 4) L. 262 should be “...while an increase of...”
- 5) L. 263-264: delete the first instance of “in larger vesicles”
- 6) L. 272 and 275: replace “emerged” with “resulted in”
- 7) L. 278: please cite a reference for “as expected”
- 8) L. 283: delete “a”
- 9) L. 292: should be “therein”
- 10) L. 417-418: should be ...” which maintained the Min oscillations in a pronounced...”
- 11) L. 437: delete “on”
- 12) L. 517: should be “identifying”
- 13) L. 573: replace “notorious” with “daunting”
- 14) L. 728 should be “chamber”
- 15) L. 747 should be “the vesicle periphery”

We thank the reviewers for their critical but valuable comments, which helped us to improve our manuscript further. After careful consideration of the main lines of criticism, we made a number of major changes, in particular:

- We took better care to emphasize the novelty of our study compared to similar reconstitution studies, but also to own prior work
- We moved all confirmatory control experiments to the supplement to reduce the bulkiness of the text and better focus on truly novel insights.
- Regarding the positive feedback mechanism between FtsZ-ring formation and pole-to-pole oscillations of Min proteins, we performed additional experiments and analysis to make our statements clearer. We also addressed in more detail the role of MinC.

By emphasizing those major points and fixing minor issues as suggested, we do believe our manuscript has greatly improved in clarity and impact. We hope that the reviewers agree.

Before answering the reviewers' comments point by point, we would like to address the major points of criticism more generally.

1) Novelty of our study and comparison with other *in vitro* reconstitution studies

We are well aware of previous studies by the Danelon, Doi, and Ueda groups using cell-free expression to reconstitute the bacterial division machinery *in vitro*, and also purified protein-based reconstitution studies, e.g. by the Erickson group and our lab. As we initially focused on results with direct relevance for our experiments and results, we apologize for any incomplete discussion or referencing. We considered it obvious that completing the full reconstitution of a minimal bacterial division ring placement *in vitro* within lipid vesicles provides a clear step ahead from what has been accomplished by our and other groups to date. We have revised the manuscript adding a more detailed comparison with those studies and more references to the text. In following paragraphs, we will detail on the novelty and impact of the presented results beyond previous studies for the more critical reviewers in particular.

Danelon and Doi groups (Godino et al., 2019, Nat. Comm., Yoshida et al., 2019, Chem. Sci.) have succeeded to express MinDE proteins, including Min wave generation, within lipid vesicles/droplets with purified GFP-MinC, probing that cell-free expressed Min proteins work as expected. Godino et al. have also expressed MinC/MinDE/FtsA proteins separately in different test tubes, mixed, and added them on supported lipid bilayers (SLBs) together with purified FtsZ to generate Min waves able to drive the FtsZ filaments. A follow-up study by the Danelon group (Godino et al., 2020, Comm. Biol.) and a similar study from Ueda group (Furusato et al., 2018, ACS Syn. Biol.), achieved the assembly of FtsZ and other FtsZ-related proteins via cell-free expression inside lipid vesicles, showing membrane deformation of lipid vesicles or "necks" by FtsZ constriction.

Although these studies have provided a plausible path towards the reconstitution of bacterial division machinery *in vitro*, and indeed succeeded in parts, none of them has achieved the co-reconstitution of full MinCDE and FtsZ-FtsA systems inside lipid vesicles so far. Therefore, the present study, where we were able to express all 5 essential components (MinCDE and FtsA-FtsZ) and witness the mutual interplay of both proteins systems inside lipid vesicles constitutes a major experimental and conceptual advance. Beyond being able to successfully express those proteins, we visualized in real time the emergence of their self-organization within lipid vesicles, providing novel insights into their complex behavior. In particular, the rapid branching of the FtsZ filaments depending on FtsA concentration on the membrane has never been reported before (Supplementary Fig.5d, e, and Supplementary Movie 3). Even more obviously, the re-organization of the FtsZ mesh into a ring structure governed by Min waves inside lipid vesicles was firstly shown in this study (Fig. 3c, d, Fig 4b-d and Supplementary Movie 7 and 8). The technical advances presented here with the cell-free system to express all five essential components has been extremely challenging, which is presumably why it has not been achieved before. Hence, without any doubt our results mark a major breakthrough

towards the ultimate goal of bottom-up synthetic biology, i.e., constructing a minimal cell from scratch.

With regard to the reconstitution of purified proteins, there are several previous studies that achieved the co-reconstitution of the FtsZ-FtsA/FtsZ-Venus-mts and the MinCDE systems. The first of these studies indeed came from our own lab (Zieske and Schwille, 2014, eLife). In this work, MinCDE proteins were co-reconstituted together with FtsZ-Venus-mts using SLBs on a rod-shaped PDMS chambers to show the geometrical effect on the MinCDE oscillations and the re-organization of FtsZ filaments towards the middle of the container. However, in the artificial environment of open PDMS chambers, the reconstitution of closed FtsZ-ring structures was not possible. Further studies demonstrated the displacement of FtsZ filaments by MinCDE waves on SLBs (Martos et al., 2015, Biophys. J., Ramm et al., 2018, Nat. Comm., Godino et al., 2019, Nat. Comm.) or inside lipid microdroplets (Zieske et al. 2016 Angew. Chem. Int. Ed.) to address the molecular dynamics of FtsZ organization. However, these experimental setups lack some critical features as a cell model, such as cell-sized confinement and deformable membrane, which appeared to be critical to observe the formation of FtsZ-ring and even constriction of lipid vesicles by the FtsZ-ring.

Concerning the independent reconstitution of FtsZ-Venus-mts alone and the generation of vesicle constriction, the creation of FtsZ-Venus-mts ring variant was firstly reported by the Erickson group in 2008, describing major membrane deformations inside tubular vesicles (Osawa et al., 2008, Science). Later, this study has been further expanded by their and our group towards the formation of necks and membrane protrusions inside and outside lipid vesicles (Osawa and Erickson, 2013, PNAS, Ganzinger et al., 2020, Angew. Chem. Int. Ed., Ramirez-Diaz et. al., 2021, Nat. comm.). Despite all the efforts, none of these studies have ever observed the assembly of a defined FtsZ-ring structure promoted by MinCDE oscillations, especially at such a large ring sizes as observed here.

The assembly of FtsZ-ring structures is challenging due to the poor understanding of how to control the Min and Fts protein machineries inside lipid vesicles. Only the extensive investigation on their molecular dynamics under macromolecular crowding conditions enabled us to reconstitute their complex dynamics, even by using the cell-free expression system. Moreover, we found the effective assembly of FtsZ-ring structures under crowding conditions together with a significant increase of pole-to-pole oscillation of Min waves (Fig. 2c-g and Supplementary Fig. 3), suggesting a positive-feedback mechanism between the Min wave dynamics and FtsZ structure. These findings could be a starting point for further theoretical considerations of biological pattern-forming systems (Please see the next comment and the answers to reviewer #1, Q4-6).

2) Positive feedback between FtsZ-ring formation and pole-to-pole oscillation of Min waves

Following the reviewers' advice, we performed additional experiments for a better characterization of the positive feedback between the pole-to-pole oscillation of Min waves and the formation of FtsZ-ring, as suggested by our study. The new results can be found in Fig1e and Fig. 2e-g. The motivation of these experiments was to answer two concrete questions; [1] whether the formation of a FtsZ-ring is directly caused by the interaction between FtsZ and MinC, and [2] whether the enrichment of pole-to-pole oscillations was already due to the presence of FtsZ bundles on the membrane or needed a defined FtsZ-ring. To this end, we tested the mutant MinC^{G10D}, which is unable to interact with FtsZ, and also the full absence of MinC in the system to decouple the causality of formation of the FtsZ-rings from pole-to-pole oscillation of Min waves from the presence of FtsZ bundles assembling a mesh.

First, we statistically analyzed the Min wave patterns in the absence of FtsZ and found that the substitution of MinC by MinC^{G10D} does not dramatically modify the occurrence of Min wave patterns at high crowding conditions (Fig. 1e). In contrast, we observed drastic differences in FtsZ-ring formation among MinC, MinC^{G10D}, or absence of MinC in the co-reconstitution assay

(42%, 10%, and 12%, respectively) (Fig. 2f). These results showed a much lower efficiency of FtsZ organization in case of using MinC^{G10D} mutant or absence of MinC, which indicated that MinC is an essential functional module for division ring placement (Fig. 2f). Strikingly, we found residual FtsZ-ring formation using MinC^{G10D} mutant or even without MinC, although efficiency stayed low (12% and 10%, respectively), suggesting that the non-specific diffusiophoresis phenomenon, by which membrane-bound objects are directionally driven by the Min proteins may be at play here (Ramm et al., 2018, Nat. Comm., Ramm et al. 2021, Nat. Phys.). This kind of physical displacement of molecules can even influence large-scale cytoskeleton organization processes inside cellular compartments, which is an exciting finding that warrants more future attention.

We further analyzed the changes of Min wave patterns, especially pole-to-pole oscillations, among different experimental conditions, such as difference of MinC and the presence of FtsZ. Intriguingly, both MinC and MinC^{G10D} increased the frequency of pole-to-pole oscillations in the presence of FtsZ, suggesting that FtsZ bundles on the membrane might change the mode selection dynamics of Min waves. Despite this overall enhancement of pole-to-pole oscillations, the efficiency of enhancement was more apparent in the case of wild-type MinC, where we observed ~20% (From 20% to 40%) increase of pole-to-pole oscillations together with high occurrence of FtsZ-rings (42%), whereas MinC^{G10D} yielded only 6% (From 25% to 31%) difference without strong enhancement of FtsZ-ring formation (10%) (Fig. 2f and g). Hence, we concluded that FtsZ bundles on the membrane might be able to enhance the pole-to-pole oscillation by affecting their molecular dynamics. However, it is still plausible to suggest that the formation of FtsZ-rings is one of the main factors to enhance the pole-to-pole oscillations by a positive feedback mechanism as we proposed in the main manuscript.

More detailed information about the additional experiments can be found in the point-to-point answers to reviewer #1, Q4-6, where we discussed more about the additional results.

Taken together, both improvements in revision and additional experiments together with original data convinced us that our study is not only expanding the technical limitations in bottom-up synthetic biology, but also providing noticeable advances in our general approach of *in vitro* reconstitution of a minimal bacterial division system, in order to disentangle the complex interplay between functional machineries in biological systems. We would be glad if the reviewers and editors agreed to our assessment and considered the study ready for publication.

Reviewer #1 (Remarks to the Author):

In this technically advanced study, the authors reconstitute parts of the bacterial cell division machinery in liposomes. They combine the MinD/C/E system with FtsZ/A using either purified proteins or *in vitro* expressed proteins. They observe various dynamical spatiotemporal organizations of these proteins on the membrane, among them central FtsZ ring formation and MinC oscillations. They empirically optimize conditions, including exploring effects of crowding agents, with the aim to observe more robust central FtsZ ring formation in the liposomes. An interesting result is that liposomes deform as central FtsZ rings seem to start contracting when the complete system is put together, going beyond previously observed less spatially controlled liposome deformations. This work represents a technical advance over previous similar reconstitutions that were either simpler in complexity or did not show MinCDE-dependent central FtsZ ring formation as convincingly as here. Surprisingly, the authors do not explain very much in mechanistic terms why the optimal conditions they find are optimal, although these systems have been studied quite extensively (separately) in the past and are at least to a certain extent well understood. Therefore, it appears that the focus of the work is not to gain new mechanistic insight into the bacterial cell division apparatus, but rather to push the limits of what's currently technically possible with respect to reconstituting a challenging *in vitro* system displaying interesting dynamical behavior, in part similar to some aspects of bacterial cell division.

We thank the reviewer for his/her feedbacks on our study. We agree that our work advances the limits of the current technical possibilities for the *in vitro* reconstitution using cell models, however, we also believe that our study provides new insights for the bacterial cell division process and the orchestration of complex systems *in vitro* (please see the general comments for all reviewers). We have addressed all the concerns below, including major changes in the manuscript, together with new experiments to give more mechanistic insight into MinCDE/FtsZ-FtsA dynamics, and hope that the reviewer will agree to them.

Major concerns/questions/suggestions:

1. It is often not clear which concentrations of purified proteins are used in the experiments and which conditions are used for the *in vitro* expressions. To allow reproducing these experiments, the critical information defining the specific conditions shown (e.g. protein concentrations) should be clearly stated in the legends to complement the otherwise detailed information provided in the Methods.

We apologize the confusions by not clearly indicating the experimental conditions in the figure legends. We have included detailed information in the figure legends to complement the information specified in the Methods.

2. Fig. 1c: Is this phase space expected given the current knowledge of the MinC/D/E system? If yes, please explain why (including references). If no, can a mechanism be proposed?

Considering the molecular mechanism of the MinCDE system, the phase diagram shown in Fig 1c (Now placed in Supplementary Fig 2c) is expected and in good agreement with a previous study (Kohyama et al., 2019, eLife, Kohyama et al., 2020, Nanoscale), where MinDE-based phase diagrams were confirmed both theoretically and experimentally using water-in-oil droplets. Briefly, in the presence of relatively higher concentrations of MinE over MinD, MinD (and MinC localization as an indicator of MinD) is mostly removed from the membrane and therefore locates in the lumen as observed in Supplementary Fig. 2c and d. In the opposite scenario, with relatively lower concentrations of MinE, MinD homogeneously binds on the membrane and does not emerge any dynamic/static patterns. Following these molecular dynamics, balanced protein ratios of MinD/E concentration would be required to favor the generation of dynamic patterns as travelling waves, pulsing or triggering waves on the

membrane (framed as dynamic waves here to simplify the outcome). Our data is therefore in good agreement with the MinCDE molecular mechanism.

In addition, we would like to highlight that this phase-diagram was firstly shown with MinD, MinE and mScarlet-I-MinC proteins (before it was observed only with fluorescently-labelled MinDE proteins) and also within the lipid vesicles (the previous study used water-in-oil droplets as experimental setup). Thus, we believe that the result presented here assures that Min wave dynamics also emerge as expected under slightly different membrane environments with more components of the system.

3. Suppl. Fig. 2c: Are ever FtsZ rings observed in the absence of MinC/D/E? If not, that should be stated for clarity. If yes, this should be included in the quantification.

We could not find any clear ring-like structures in the absence of MinCDE proteins inside lipid vesicles regardless of the macromolecular crowding concentration. Therefore, together with the additional experiments (Please see next comments), Min(C)DE wave is strictly necessary to form FtsZ-ring structure in lipid vesicles. The figure is now placed in Fig. 1c. It has been stated in the manuscript (l. 195-196) and legend of Fig. 1 for clarity.

4. Fig. 2c: FtsZ rings form in the presence of MinC/D/E. Has it really been demonstrated that FtsZ ring formation necessarily requires MinC oscillations? For example, can it be excluded that simply stronger bundling of FtsZ filaments due to the simple presence of MinC/D/E leads to ring formation? Similarly: is centering of the FtsZ ring really due to the oscillations or does the ring simply form where the diameter is largest to minimise the bending energy of the filaments? The authors probably have the answers and could state them explicitly.

We would like to thank the reviewer for this insightful feedback, and hope this question is adequately addressed by our additional experiments, where we analyzed the frequency of FtsZ structures inside vesicles with MinDE proteins but without the presence of MinC or by using an FtsZ-interaction defective MinC mutant (MinC^{G10D}) (Hu et al., 1999, PNAS, Zieske and Schwille, 2014, eLife) (Fig. 2e and f). In principle, we could expect the lack of ring-like structures in the absence of MinC, which depolymerize FtsZ filaments on the membrane. However, we have observed the formation of FtsZ-ring structures either in the absence of MinC or by using a MinC^{G10D} with a similar frequency ~10-12%, much lower than when MinC is present (~42%) (Fig. 2f).

Interestingly, these results are perfectly matched with the previous observations on Min waves *in vitro*. Firstly, our recent study showed that FtsZ is displaced by the Min waves on SLBs and their localization is significantly sharpened by the participation of MinC in the Min wave, involving a direct interaction between FtsZ and MinC (Ramm et al., 2018, Nat. Comm.). It has been also demonstrated that FtsZ filaments can be re-organized by the MinCDE waves in other *in vitro* systems (Zieske and Schwille, 2014, eLife, Martos et al., 2015, Biophys. J., Godino et al. 2019, Nat. Comm. Biol.). Therefore, it makes much sense to expect clearly higher efficiency of formation of the FtsZ-ring structures in the presence of MinC within our experimental system.

On the other hand, our previous work (Ramm et al., 2018, Nat. Comm., Ramm et al. 2021, Nat. Phys.) also found that the MinDE waves are able to organize membrane binding molecules by a density-dependent physical displacement mechanism without involving any specific interaction. This non-specific process based on diffusiophoresis that has never been described in cell biology could represent a generic regulation system for intracellular spatiotemporal organization. Hence, this together with our new data convinced us that FtsZ-ring structures could be observed in the absence of MinC or by using MinC^{G10D}, even though the efficiency of the regulation decreases (Fig. 2e and f).

Taken together, we conclude that the presence of MinC is not strictly necessary for the assembly of an FtsZ-ring structure, however, drastically enhances FtsZ regulation and thus is an essential factor for cell-division both *in vivo* and *in vitro*. Moreover, we believe that our new data demonstrates the power of the recently discovered diffusiophoretic mechanism, taking place even in lipid vesicles and on assembling filaments, which may catalyze new research on its potential role and relevance for cellular organization.

5. Similar question concerning the increased frequency of MinC oscillations in the presence of FtsZ: could it simply be the presence of FtsZ instead of an FtsZ ring that promotes MinC oscillations?

We thank the reviewer again for this interesting question. According to our results, we can assure that the presence of higher numbers of FtsZ bundles on the membrane enhances the emergence of MinCDE pole-to-pole oscillations, as demonstrated in the data shown in Fig. 1c, e and Fig. 2f and g. However, the presence of FtsZ-ring structures on the membrane cannot be decoupled from the higher number of bundles, rendering it difficult to assure that the presence of discernible FtsZ-rings is indeed required for the enhanced occurrence of pole-to-pole oscillations (although it makes sense to consider them as one of the major reasons (see next answer)).

Therefore, we have performed additional experiments analyzing the MinCDE waves using MinC^{G10D} unable to interact with FtsZ. MinC^{G10D} is less able to form FtsZ-ring structures on the membrane, as compared to wild type MinC (Fig. 2e and f), which allowed us to partially decouple the presence of bundles and the formation of FtsZ-rings. In this case, we have only analyzed the crowding condition that has shown higher numbers of FtsZ-rings (100 g/L Dextran70). As shown in Fig. 1e, the substitution of MinC by MinC^{G10D} does not have a significant impact on the frequencies found for the different MinCDE patterns. Interestingly, the presence of FtsZ and high crowding conditions increased the Min pole-to-pole oscillations less than using MinC. We observed an increase of ~20% (From ~20% to ~40%) in the presence of MinC while using MinC^{G10D} it was only ~6% (From ~25% to ~31%) (Fig. 2g). Although the frequencies are not dramatically different, our results suggest that the presence of a higher number of FtsZ-rings inside vesicles is able to enhance the emergence of pole-to-pole oscillations. Hence, our additional experiments allow us to demonstrate that FtsZ bundles on the membrane enhance the emergence of Min oscillations, the formation of FtsZ-rings being one of the main factors involved in the process, improving the enhancement of pole-to-pole oscillations even further.

6. The authors suggest that there are mutual positive feedbacks between FtsZ ring formation and MinC/D/E oscillations. Is this expected based on the current understanding of these systems from previous work? If yes, that should be explained. If not, can the authors propose a mechanism?

The positive feedback suggested in this study is not expected by the current understanding of the molecular dynamics of Min waves, however, it is plausible to assume that this phenomenon is closely related to the geometrical effects on Min waves as described earlier. In our previous studies (Schweizer et al., 2012, PNAS,) and also by others (Wu et al., 2015, Nat. Nanotech.), it has been reported that Min proteins can sense the geometry of the accessible membrane area and re-organize the wave patterns. For example, in the simplest case, Min proteins were confined in the rod-shaped PDMS chamber and therefore, appearing only as the pole-to-pole oscillation (Zieske and Schwill, 2014, eLife). Moreover, by introducing the “septum” geometry in this rod-shaped chamber, Min waves switched their dynamic patterns from single to double nodes oscillation, giving direct evidence of the sensitivity of Min waves to physical barriers on the membrane. In our additional experiments, we have observed a higher occurrence of pole-to-pole oscillations for the case of MinC in comparison with MinC^{G10D} which promotes less the formation of FtsZ-rings. This result points towards a positive feedback, in which more pronounced FtsZ-rings stabilize and promote pole-to-pole oscillations.

Our observations together with the studies referenced above suggest that Min waves could be also potentially affected by the presence of molecules that serve as physical barriers on the membrane. Considering the presence of thick FtsZ bundles on the membrane, they might behave as a physical barrier that may still be pushed by the Min wave, but at the same time also separate the distribution of Min proteins into two distinct regions inside the lipid vesicles. Assuming this “separator” effect, Min waves likely cannot access the successive membrane area required to emerge traveling wave. Thus, it is not surprising to observe a certain response from the Min waves to the presence of the FtsZ bundles on the membrane. However, our results are only the first observation of this phenomenon and more experiments, together with a theoretical model, will be required to confirm a positive feedback also quantitatively.

In addition, in case of cell-free system, a mild constriction of lipid vesicles breaks the symmetry of the sphere and therefore may reinforce the pole-to-pole oscillation. This trend was also confirmed among the PDMS open chambers in our previous studies (Zieske and Schwille, 2013, *Angew. Chem. Int. Ed.*, Zieske and Schwille, 2014, *eLife*), suggesting the geometry sensing by Min proteins preserved across the different experimental platforms.

7. Do the authors know how protein concentrations for the various proteins develop over time in the in vitro expression experiments? Do the observations in these experiments agree with those in the experiments with purified proteins, given the development of the protein concentrations over time?

Indeed, there are some techniques that have been used to measure expressed protein concentrations in cell-free systems, such as fluorescently or radioactively labelled amino acids (Furusato et al., 2018, *ACS syn. biol.*), and tag-based organoarsenic molecules (ReAsH, Wick et al., 2019, *ACS syn. Biol.*). However, in this scheme it is technically challenging to bring these techniques to the measurements inside lipid vesicles for several reasons, especially in case of the expression of multiple proteins at the same time. For instance, it is never possible to distinguish the fluorescence from freely-diffusing dyes and synthesized proteins. Also, if multiple proteins are expressed at the same time, labelled amino acids would be incorporated into all kind of proteins and therefore, fluorescence among different proteins cannot be distinguished. Moreover, due to the small choices of the available biologically benign fluorescent molecules, it seems also quite difficult to label the multiple proteins at the same time, especially in the case for organoarsenic molecules.

Due to these difficulties, they have been used mostly for the assays in test tubes, also including mass spectroscopy measurements (Godino et al. 2019, *Nat. Comm.*). However, since cell-free expression dynamics are quite different between lipid vesicles and test tubes (Yoshida et al., 2019, *Chem. Sci.*, Sakamoto et al., 2018, *Sci. Rep.*), the measurements in a test tube don't give very informative results for our cell-free experiments inside lipid vesicles. Additionally, it might be possible to measure the protein level by correcting the lipid vesicles after certain duration of expression, however, it would be extremely difficult to measure the expression level of proteins in the individual vesicles. Of course, in this case, the changes of the protein concentration along with the expression is not measurable, although it might be quite important to know protein dynamics depending on the concentration.

The only possible method might be the expression of fluorescence proteins, as we used to determine the effects of macromolecular crowders in Supplementary Fig. 4. However, it is also technically difficult to express and detect the 5 different color fluorescent proteins at the same time, considering that a gain of the molecular weight might decrease the expression level, and also cross-talking of the fluorescent signals. Moreover, both Min and Fts proteins are heterogeneously assembled on the membrane, it would be adding an extra layer of complexity to measure the correct protein concentration inside lipid vesicles. Therefore, we concluded that it would be quite challenging to measure the exact concentrations of the expressed proteins in our system, being beyond the scope of our study.

8. Fig. 3: FtsZ ring formation is less stable here, as the authors observe, which may be the consequence of changing protein concentrations over time. Based on the current understanding of these dynamical systems, can a mechanistic explanation be proposed?

As shown in Supplementary Fig. 2c and d for the purified proteins and Supplementary Fig. 6b for the PURE expression, the dynamic behavior of Min patterns is highly dependent on the MinDE concentrations and their ratio. Similar trends of wave dynamics could also be found in the literature (Kohyama et al., 2019, eLife, Yoshida et al., 2019, Chem. Sci., Takada et al., 2022, Sci. Adv.). These differences in the available patterns are explained by the change of MinDE concentration over time by cell-free expression, and indeed, Supplementary Fig. 6b shows that wave patterns tend to become traveling waves rather than pole-to-pole oscillations. Thus, it is plausible to suggest that Min wave patterns depending on MinDE concentrations is a decisive factor to form the FtsZ-ring structure.

Although the MinDE ratio is an important factor, FtsA concentration might also affect the stability of the FtsZ structure on the membrane. As shown in Supplementary Fig. 5b, c, and Supplementary movie 3, FtsZ structures became more branched and flexible filaments upon FtsA expression, rather than thicker and straighter filaments in the early phase of the cell-free experiments. Additionally, it was previously described that FtsA decreases the lateral interactions of FtsZ, which could also decrease the stability of FtsZ filaments (Krupka et al., 2017, Nat. Comm.). This fact suggests that the FtsZ:FtsA ratio affects the stability of FtsZ structure on the membrane, becoming more unstable at higher FtsA concentration than FtsZ. Therefore, since FtsZ concentration was fixed at 2 μ M, while FtsA and MinDE concentration increases over time (Fig. 3c), a relatively low amount of FtsZ compared to FtsA might result in the formation of less stable FtsZ-ring like structure in lipid vesicles.

9. Fig. 5: Why are in these experiments multiple FtsZ rings observed, whereas in experiments without FtsA single rings could be seen?

Firstly, it is important to mention that we do not consider these structures as multiple rings but rather the condensation of multiple bundles in the equatorial plane of the vesicle. The differences observed between expressed and purified system are intriguing, although they may not be systematic and might be explained by considering the features of each system.

The most noticeable difference between both systems is the membrane binding process of FtsZ, where FtsZ-Venus-mts binds directly to the membrane through its membrane targeting sequence (mts) region, while FtsZ-G55-Venus-Q56 is attached through FtsA in the cell-free expression system. Notably, we demonstrated that FtsZ-G55-Venus-Q56 is an active protein with similar GTPase activity and able to assemble similar filaments and bundles to the FtsZ-wt protein on SLBs (Supplementary Fig. 6f and g). Regarding these differences on the membrane binding nature of FtsZ, it has been stated that different attachment to the membrane affects the dynamics of FtsZ on the membranes (Loose and Mitchison, 2014, Nat. Cell Biol., Ramirez-Diaz, 2018, Plos. Biol., Garcia-Soriano et al., 2020, Sci. Rep.). The fact that FtsA decreases the lateral interactions of FtsZ (Krupka et al., 2017, Nat. Comm.) also suggest the less effective condensation of FtsZ filaments. Moreover, it was expected to have higher concentrations of FtsZ in the expression system which might explain the high number of FtsZ filaments observed in the cell-free system. In addition, the buffer composition required by the PURE expression system contains different components such as higher magnesium (see the answer for the question no. 10 from reviewer #3) among others, which might have a high impact over the behavior of the proteins systems. Thus, together with the differences in the protein concentration among the systems, intrinsic differences of the FtsZ membrane binding motif might account for the different molecular dynamics observed in the distinct systems.

However, more experiments would be needed to fully understand the differences between both systems, such as testing the different membrane binding strength of FtsZ by substituting the

mts region with different attachments (Garcia-Soriano et al., 2020, Sci. Rep). In addition, the use of other membrane attachment components such as ZipA might be interesting to study in the cell-free system, although both of them are far out of the scope of this study. We would like to thank the reviewer for this insightful suggestion and keep the question on our radar in future projects.

10. Why are larger liposomes more strongly deformed by FtsZ rings than smaller liposomes. Which biophysical parameter could influence this deformability?

As outlined in the discussion, we assumed that the expressed protein concentration in bigger vesicles was higher than in smaller ones. This is not an implausible assumption, as low surface-to-volume ratios are generally preferable for all processes that occur in solution, such as our cell-free expression. To prove this, we measured the concentration of FtsZ on the membrane and confirmed a good correlation between vesicle size and the concentration of FtsZ on the membrane (Supplementary Fig. 5d, e, 6h, and i). Therefore, FtsZ may be able to generate larger forces at higher concentrations, and thus more readily induce membrane transformations.

Similarly, the concentration of FtsA might be higher in the bigger vesicles. Since we previously reported that membrane attachment of MinDE proteins through their amphipathic helices induces an imbalance in membrane curvature in lipid vesicles (Litschel et al., 2018, Angew. Chem. Int. Ed.), it likely also happens with FtsA and might induce larger scale transformations within bigger vesicles. To further investigate the deformation of lipid vesicles, addition of transmembrane proteins, such as ZipA that is also a part of the *E. coli* divisome but only poorly understood in experiments so far (Furusato et al., 2018, ACS syn. Biol.), would give more insights into the local bending of the membrane in the reconstituted system. Another question that warrants more in-depth consideration, but goes beyond the scope of this manuscript.

Regarding membrane properties, in spite of its intuitive plausibility, we could not find any evidences in literature that proves generally larger membrane deformability in larger vesicles. Rather, we only found that there is no significant difference on mechanical properties among GUVs (Schäfer et al., 2013, Langmuir). However, since in this paper relatively small GUVs (2-9 μm) were studied compared to efficiently deformed vesicles by FtsZ-ring (larger than 24 μm), it is still not clear if there are any contributions to membrane deformability based on the vesicle size, however, it would be an intriguing question to further address in the future studies by applying biophysical measurements on the membrane softness such as flicker microscopy (Salinas-Almaguer et al., 2022, Sci. Rep.) or compression by AFM (Schäfer et al., 2013, Langmuir, Schäfer et al., 2015, Soft Matter).

Minor comments:

1. Line 48: "low surface-to-volume ratio" - compared to what? Some people might think that the ratio is high, depending on which comparison they have in mind.

We added "compared to flat SLBs" (l. 47) to clarify the point.

2. Line101: It is not clearly stated from the start that purified proteins are used for the first experiments.

To specify the proteins that we use for the first experiments, we revised the according sentence from "MinCDE self-organization" to "Min waves using MinD, MinE, and mScarlet-I-MinC" (l. 130-131)

3. For all figures: Often it is not stated at which times after start of assembly are the displayed microscopy images are taken, which appears however important as the studied systems develop over time.

We usually started the observation after 10 min of the preparation of lipid vesicles. We added this information in the figure legends and methods section.

4. Fig. 1b. To make it easy, it could be stated in the figure which protein is shown. For the non-expert, it should be explained how the kymographs are generated: 1D lines are shown over time, generated from a 3D object. The presented microscopy images are rather small.

We specified the proteins we visualized for microscopy observations in all figures. For the generation of kymographs, we added further detailed explanation in the legend of Fig. 1. We also added some enlarged images of FtsZ-ring within lipid vesicles such as Fig. 2c, e, 3d and 4c.

5. Fig. 1c. Example data for the 3 categories could be shown, at least in the Supplement.

We added Supplementary Fig. 2a to show the dynamics of the different Min wave modes in the two-phase diagram found in our experimental setup. We also added a brief explanation of each Min mode in the legend.

6. Citations. Line 166. Reference 50 is a bit odd, because a review is called to support a rather detailed technical statement. Was this intended? Same for other reviews that are cited: can original literature be cited more specifically?

We are sorry for this incorrect citation here, the manuscript we have intended to mention was the following: Martos et al. 2012 "Isolation, characterization and lipid-binding properties of the recalcitrant FtsA division protein from Escherichia coli." PLoS ONE 7, e39829. In which it is described an optimized method to purify FtsA, a challenging protein due to its recalcitrant nature. According the revision of the original main manuscript, this citation and regarding results have been moved to Supplementary results (Supplementary file, l. 133-135). In addition, we have checked the manuscript thoroughly and found some inappropriate citations for the technical statements. Therefore, we now changed them to the correct citations.

7. Fig. 2e. Please show some representative original image data for the different categories.

We added Supplementary Fig. 3 to show the time-lapse images of different categories of FtsZ structures found in presence of the Min oscillations.

8. Line 240: Instead of only stating that "division ring placement" has not yet been achieved, it would be useful to also say what has been achieved so that the advance here is clear.

We further stated what has been achieved in the past studies in l. 272-279.

9. Line 420: Can the evidence for "constriction" (in 2 dimensions) instead of "squashing" (deformation in only 1 dimension) be presented?

We have added the rotational views of the FtsZ-ring structure within a constricted vesicle to show the 2-dimensional constriction by FtsZ-ring structure rather than 1-dimensional squishing in Supplementary Fig. 7d and Supplementary movie 9.

10. Line 428: what does "some sort of dissipative mechanism" mean? A more precise statement could be clearer.

We are sorry for the unprecise statement, we have edited the manuscript accordingly: "Surprisingly, although we suspect that some sort of energy-driven dynamic mechanism is at place in these vesicles, the spatial regulation of the FtsZ-ring like structure to the middle of the

vesicles seems to be preserved without pole-to-pole oscillation of Min waves.” (Supplementary file, l. 215-218)

11. Line 440: "radically" - really? A more objective way to present the result would be to just state the quantity of the change.

We would like to thank the reviewer for this correction. We added the quantitative comparison of the percentage of deformed vesicles (l. 418) to improve clarity.

12. Line 485: why is it so "unexpected" that FtsZ can cause constriction? Based on what? It is implied that constriction is rather expected for smaller liposomes. Based on which mechanistic argument?

It was surprising as previous studies found that treadmilling and GTP hydrolysis of FtsZ-mts can only cause deformations on the pN range (1-2pN) (Ramirez-Diaz et. al., 2021, Nat. comm.), while a bending force of at least 10-15 pN would be required to cause minor deformations in large compartments such as Erythrocytes (Henon et al. 1999, Biophysical Journal), which are highly comparable to lipid vesicles in isosmotic conditions. Interestingly, it has also been estimated that a force of around 8 pN is needed to fulfill division in *e. coli* cells (Lan et al. 2007, PNAS). In our case, we have only observed membrane deformation using the cell-free expressed system, suggesting that higher bending forces are being exerted by the FtsZ-FtsA system in contrast to the FtsZ-mts protein, generating effective constriction forces even under isosmotic conditions. We are grateful to the reviewer for a constructive comment and the additional discussion has been stated in the manuscript (l. 508-518).

13. Some jargon/abbreviations could be explained/introduced for the non-expert: e.g. SLBs, sfGFP,

We revised the manuscript to introduce explanations for SLBs (l. 39), sfGFP (l. 587), Venus and mts (l. 114-115).

Reviewer #2 (Remarks to the Author):

The manuscript titled “In vitro assembly, positioning and contraction of a division ring in minimal cells” by Kohyama and co-authors reports the reconstitution of a cellular division mechanism comprised of FtsZ/FtsA and MinEDC cell division proteins either purified or cell-free synthesized and encapsulated inside liposomes of sizes comparable to living cells. The MinEDG system exhibits oscillations inside the liposomes and a slight contraction of the membrane of liposomes is observed due to the assembly of FtsZ/FtsA rings. Min-wave assisted FtsZ-ring assembly within liposomes is observed by two different approaches, (1) one with purified proteins and (2) one using a cell-free expression system, the PURE system. The importance of macromolecular crowding in solution during the process of FtsZ ring formation is observed and discussed in the case of purified proteins. The occurrence of FtsZ rings increases with the concentration of crowders (here Ficoll and Dextran) with purified proteins. When proteins are synthesized using the PURE cell-free system, similar results are obtained. Reconstituting molecular mechanisms in cell-free conditions enables to separate their biochemical and biophysical properties from the complexity of real living cells. This approach has become quite popular in bioengineering and constructive biology. Reconstituting a cell division mechanism, such as the one found in bacteria, is especially important as it is a critical step towards building artificial cell systems, an even greater undertaking for which major research programs have been launched on different continents. The experiments are in general well done and well described. That said, the manuscript has several major issues, the two major ones being that (1) this work is very close to an article published recently and consequently it does not seem to bring substantial new information, and (2) the manuscript is often confusing as it lacks a clear message about its novelty.

We thank the reviewer for the valuable feedbacks and concerns, although we were surprised by the apparent misunderstanding about the novelty of our study. We have addressed the concerns in our major comments for all reviewers and also some individual comments. At the same time, we have carefully checked and revised the manuscript to improve the coherence and clarity of our messages.

Major concerns:

- the work presented here is very close to the studies published recently by Godino et al 2019. It is difficult to understand where the novelty is. Phenomenologically, there is nothing really new compared to what Godino et al 2019 have published.

We appreciate the concern, however, we respectfully disagree with this statement, and indeed we have specified the novelty of our study in the summary for all reviewers above. We believe that our work provides new insights into the molecular system and sufficient technical novelty for the *in vitro* reconstitution of a minimal cell division system far beyond the studies published earlier. We hope the reviewer will agree with our major comments about the novelty of current study.

- the manuscript is often confusing, as there is no clear path in the setup of the experiments. The number of experiments presented is too large, which dilutes the message. Trying to publish a scientific study does not consist of flooding a manuscript with a load of experiments hoping that it looks good and impressive. Yes, a lot of work is presented in this manuscript, but it lacks some coherence. The manuscript should be shortened, the experiments condensed to gain clarity, and the novelty of the work should be clearly explained.

We are sorry for the bulky original manuscript which might have been distracting for readers, even though we believe those many experimental evidences themselves had been essential to lead us to the major breakthroughs (as we believe) in the reconstitution of such complex biological systems. Following this comment, we have thoroughly revised the manuscript and omitted some confirmatory parts of the experiments as well as related paragraphs from the main manuscript and now they have been moved to the Supplement. We agree that this brings great improvements in conveying the clarity of messages and highlighting novelty of our study, and therefore are grateful for this suggestion.

- molecular crowding in solution is optimized and discussed only for the system based on purified proteins and not with the PURE cell-free expression system. It is known, for instance, that crowding agents can impact the processes of transcription and translation as well as the self-assembly of cytoskeleton protein. Therefore, the optimum crowding conditions could be completely different from the one observed with purified proteins. Macromolecular crowding optimization should be done with the PURE system (see other concerns), or at a minimum discussed properly when the PURE system is used.

We have already tested the effects of macromolecular crowders on cell-free expression in Supplementary Fig. 4, where we examined the expression level of sfGFP at different crowder conditions. The final concentration of macromolecular crowding for cell-free expression (50 g/L of Ficoll70) was selected considering the level of expression to produce sufficient amounts of proteins to reconstitute the Min waves, allowing also the correct bundling dynamics of FtsZ. Indeed, the optimal crowding condition found for the cell-free expression system differs from the one for the purified system. Following the frequency of FtsZ-ring formation using purified proteins, the best condition was rather 100 g/L of Dextran70 (Fig. 1c, Fig. 2f and Supplementary Fig. 1a), while it is not suitable for cell-free expression system causing a low expression level (Supplementary Fig. 4). Due to the complex features of the PURE expression system, a deeper biochemical characterization of the macromolecular conditions would again be beyond the scope of the manuscript.

- it is never clearly explained why macromolecular crowding is studied in solution and how it could affect proteins that are predominantly located/bound at the membrane.

We apologize for the lack of a detailed explanation of why macromolecular crowding in solution is beneficial for membrane-based systems. From experimental evidence, we find that macromolecular crowding has a great impact on membrane systems and can name at least three reasons:

Firstly, it is well established that macromolecular crowding in solution non-specifically enhances molecular interactions, by a volume exclusion effect that has the potential to significantly modulate the kinetics and equilibria of a large number of macromolecular reactions taking place inside the cell. They have been summarized by the formation of macromolecular complexes in solution, association-dissociation rates, compaction or folding of proteins binding of macromolecules to surface sites among others (Zhou et al., 2008, *Annu. Rev. Biophys.*, Rivas and Minton, 2022, *Annu. Rev. Biochem.*). Then, it is expected to increase the rate to transition state-limited association reactions and to decrease the rate of fast, diffusion-limited association reactions, such as FtsZ polymerization dynamics. This non-specific enhancement of protein-protein interactions might be therefore beneficial to our system.

Secondly, the effects of macromolecular crowding in solution and on the membrane are thermodynamically and kinetically linked. Eventually, any effect on the molecules in solution will affect their equilibria with the membrane, and thus all the interactions with other components of the system. The volume exclusion effect as result of macromolecular crowding might lead to an up-concentration of the protein molecules in the interphase between solution and surface, enhancing their interaction with the membrane. For example, an up-concentration of molecules enhances the condensation of FtsZ bundles on the membrane, which can eventually facilitate the assembly of ring-like structures driven by Min oscillations.

Lastly, it is often considered that membrane systems function only in 2D though there is usually a constant exchange with the protein pool found in solution. Indeed, the MinCDE and FtsZ systems are perfect examples of this phenomenon. MinCDE waves can only emerge depending on a constant exchange of Min proteins with the pool in solution, recruiting Min proteins to the membrane and dissociating from it. In the case of FtsZ, its treadmilling mechanism is also based on a constant exchange of molecules with the pool in solution, allowing their correct functionality (Ramirez-Diaz et al., 2018, *Plos. Biol.*). In both cases, we could find previous studies that have demonstrated a significant impact of volume exclusion effect from macromolecular crowders in solution. For example, crowding environments promoted the bundling, increased oligomerization or even assembling the liquid-liquid phase separation on FtsZ (Gonzalez et al., 2003, *J. Biol. Chem.*, Monterroso et al., 2016, *PLoS One*, Robles-Ramos et al., 2021, *Biochim. Biophys. Acta - Mol. Cell Res.*). On the other hand, the expansion and travelling of MinDE waves decreased their wavelength and velocity (Schweizer et al., 2012, *PNAS*, Martos et al., 2015, *Biophys. J.*), also enhancing the emergence of Min waves in microdroplets (Kohyama et al., 2019, *eLife*). Importantly, addition of macromolecular crowders seems to be required to observe FtsZ bundling inside vesicles (Cabre et al., 2013, *J. Biol. Chem.*, Osawa and Erickson, 2013, *PNAS*, Furusato et al., 2018, *ACS Syn. Biol.*, Godino et al. 2020, *Comm. Biol.*), making it an essential component for the present project.

Considering all these findings, we believe that macromolecular crowding in solution represents a promising tool to improve the *in vitro* reconstitution of protein systems either in solution or on the membrane, allowing also the mimicking of certain features found in the crowded cellular environment. However, to clarify these points, we have added a new discussion in Supplementary discussion (Supplementary file, I. 257-261 and 268-273) and hope that it is clearer in the current version.

- what is the point of presenting two approaches, one based on purified proteins and one on cell-free expression? This renders the message of the work confusing. The cell-free protein synthesis approach seems more relevant in the context of artificial cells.

We agree that the cell-free protein expression system is probably more relevant for the bottom-up construction of synthetic minimal cell models beyond a certain number of functional proteins. However, both systems offer certain advantages in the reconstitution of *in vitro* systems, and the beauty here is that the number of components is just still manageable to accomplish a true comparison and by this, get better insight in potential mechanistic differences. Especially, the purified protein system has served as a useful platform to optimize our experimental conditions, allowing the improvement of the system and tuning conditions that could be used in the cell-free system with slight differences. The PURE cell-free expression system implicates a higher degree of complexity than purified protein systems and involves higher technical challenges, rendering it difficult to tune parameters and optimize certain conditions such as salt conditions, pH, lipid composition, macromolecular crowding and protein concentrations. Using purified proteins, we could not only find the most suitable conditions for our experimental setup but also accomplished to assemble a dynamic ring-like structure in a controllable manner. On the other hand, as the reviewer points out, our cell-free system provides the advantage of constructing more complex biological systems within synthetic cells. Moreover, our visualization technique allows us to capture the real time dynamics of these biological systems, such as the emergence of Min waves and formation of FtsZ-ring structure, which could not be achieved with purified proteins due to the methodological limitations. Therefore, we strongly believe that exploring and comparing both systems is essential for a better scientific appreciation of their particular differences, and highly useful for the community that is essentially still split into cell-free and purified-protein approaches. It has been discussed in Supplementary discussion for clarity (Supplementary file, I. 234-239)

- It is also not clear how/why the five proteins were chosen. They do belong to the bacterial division mechanism. But many other proteins are involved in the Fts-ring formation process.

It is known that FtsZ is not only one of the main components of the divisome complex but the first protein to localize at midcell. Moreover, this accumulation of FtsZ can only be established through its binding to the membrane by interaction with FtsA or ZipA, resulting in the assembly of a ring-like structure called proto-ring. As ZipA is found only in gammaproteobacteria cells such as *E. coli* (Hale and de Boer, 1997, Cell, Margolin, 2000, FEMS Microbiol. Rev.), we decided to use only FtsA in this study since it is sufficient for the membrane binding of FtsZ. At the same time, MinCDE proteins are an essential system in *E. coli* cells to position the FtsZ-ring at the equatorial plane of the cell, enabling symmetric cell division. Both systems are considered part of the early components of cell division, assembling and localizing the early division ring which leads the development of the division machinery and eventually division. Considering the extensive knowledge of both protein systems (either FtsZ-FtsA or MinCDE), even studied together in some studies (Zieske and Schwillie, 2014, eLife, Ramm et al., 2018, Nat. Comm., Godino et al. 2019, Nat. Comm.), it seems obvious that building a minimal division system would require the use of at least these early components. The use of these minimal components in our work might establish the basic platform for the addition of other elements to the system in order to reconstitute the divisome and eventually fulfill division of lipid containers.

Minor concerns:

- with purified proteins, macromolecular crowding appears to be essential for Fts-ring formation. The crowders used are Dextran and Ficoll, both routinely employed *in vitro* to emulate crowding in biological solutions. PEG, the most popular and characterized crowding agent, is not mentioned. This should be discussed.

As the reviewer mentioned, PEG is one of the most known and used macromolecular crowders. However, it has been stated previously that repulsive interaction of PEG with other molecules

cannot be fully described quantitatively by an effect of excluded volume alone, involving certain attractive interactions between PEG and the hydrophobic side chains on the protein surface (Minton, 1983, Mol. Cell. Biochem., Winzor and Wills, 2006, Biophys. Chem., Zhou et al., 2008, Annu. Rev. Biophys.). The strength of this positive attraction varies among different proteins and indeed, a negative impact over the FtsZ dynamics has been demonstrated previously (Monterroso et al., 2016, PLoS One, Monterroso et al., 2016, Sci. Rep.). Monterroso et al. reported a higher tendency of FtsZ localization in either Dextran or Ficoll phase when FtsZ is encapsulated in PEG/Dex or Ficoll/PEG environments, suggesting that FtsZ avoids to locate in the PEG solution.

On the other hand, the attractive forces found in PEG is absent for other water-soluble proteins and polymers also extensively used as macromolecular crowders such as Dextran, Ficoll or BSA which non-specific interaction with protein molecules can be described using pure excluded-volume models (Minton, 1983, Mol. Cell. Biochem., Rivas et al. 2001, PNAS, Winzor and Wills, 2006, Biophys. Chem.). To date, some studies have been carried out by using Dextran/Ficoll crowders to investigate the polymerization dynamics of FtsZ (Gonzalez et al., 2003, J. Biol. Chem., Monterroso et al., 2016, Sci. Rep.). Therefore, we have decided to exclude PEG from our experimental setup to avoid any negative impact in our results. We have added new discussions in Supplementary discussion (Supplementary file, I. 261-267).

- considering that the proteins used or expressed are located at the membrane, it would have been interesting to also study the impact of molecular crowding at the membrane in the Fts-ring formation.

We thank the reviewer for the intriguing suggestion. As discussed above, we believe that our current experimental setup especially with purified proteins is adequate to investigate the effects of macromolecular crowding inside lipid vesicles. However, it is also true that there is still little knowledge regarding FtsZ polymerization dynamics on the membrane under crowding conditions. In this regard, we would expect that use of PEG-DOPE or other polymer-conjugated lipid might be a good candidate to study the impact of macromolecular crowding on the membrane, following some previous studies done by the Noireaux lab using MreB protein (Garenne and Noireaux, 2020, Biomacromolecules, Garenne et al., 2020, PNAS). However, these studies are beyond the scope of our present manuscript. We have included some comments in the supplementary discussion (Supplementary file, I. 337-342).

- Fig 3c: is it possible to estimate the concentration of proteins produced?

Please see the answer for the reviewer #1, no. 7.

- a table that summarizes the frequency of the different patterns (rings, dot ...) at different concentrations of crowding agents with purified or expressed proteins would be useful.

We thank the reviewer for the constructive feedback. According to the Q4-6 from reviewer 1, we have updated Fig. 2f to summarize the frequency of FtsZ structures. In addition, we attached a table of summary of FtsZ structures (please see below), although it is essentially the same as Fig. 2f and therefore we have not included it in the manuscript.

experimental condition	crowder	Concentration of crowder (g/L)	FtsZ structures (%)				
			Ring + deformation	FtsZ-ring	Mesh	Dots	Lumen
wt MinC	No Add.	0	0,0	0,0	2,8	80,0	17,2
	Dextran70	25	0,0	3,6	32,9	57,5	6,0
		50	0,0	10,2	39,3	39,0	11,4
MinC mutant	Dextran70	100	0,0	41,6	48,4	8,9	1,1
without MinC		100	0,0	9,5	55,4	28,5	6,6
		100	0,0	12,0	51,0	26,1	11,0
wt MinC	Ficoll70	25	0,0	1,6	11,2	76,0	11,2
		50	0,0	3,0	29,0	63,4	4,6
		100	0,0	17,1	45,9	29,3	7,7
cell-free exp.		50	28,1	25,0	46,9	0,0	0,0

- the discussion section should be shortened. It is too long and not very clear. The discussion should include just a few clear points and take-away messages.

As stated above, we have omitted several parts, mainly confirmatory control experiments, from the main manuscript that have now been moved to the supplementary discussion. We thank the reviewer for the suggestion and hope the manuscript is now easier to follow.

- it would have been interesting to determine if the lipidic composition affects the assembly and the positioning of the ring within the liposomes. This should be discussed.

We thank the reviewer for this meaningful suggestion. The study of different lipid compositions could be addressed in future experiments, as they might be able to determine the degree of deformation of the membrane and the self-assembly of the ring-like structure. In addition, it is known that the lipid composition and charge of the lipid membrane affects the behavior of the Min proteins (Vecchiareli et al., 2014, Mol. Microbiol., Kohyama et al., 2019, eLife). However, the limited knowledge of the Min dynamics inside vesicles would require an extensive study to determine the effect of membrane composition, which is therefore beyond the scope of our present manuscript. We have also discussed the potential effects of lipid compositions in supplementary discussion (Supplementary file, l. 342-347).

Reviewer #3 (Remarks to the Author):

This manuscript reports the complete reconstitution of the E. coli FtsA-FtsZ polymer system and the MinCDE spatial regulatory system in giant unilamellar vesicles, starting with Min oscillations that corral FtsZ-FtsA polymers into a medial zone, which ultimately leads to a small degree of membrane constriction at mid-liposome that may mimic the constriction forces at midcell in E. coli. The main improvements over previously reported bacterial divisome reconstitutions are the ability to synthesize all 5 proteins in a cell free system from DNA templates, following the transformation of a randomly localized FtsZ-FtsA polymer bundle within the liposome into a narrow band at the medial position over time, and the ability of this band to partially constrict the liposome membrane. Moreover, the data suggest that not only do Min oscillations spatially constrain FtsZ-FtsA to a medial band, but also the FtsZ-FtsA medial band enhances the stability of pole-to-pole Min oscillations, which can also devolve into other types of movements that are generally not observed in vivo. The frequency of liposomes exhibiting these behaviors seems to be higher in this study than in previous studies, and the effect of specific crowding agents has been optimized. Together, these findings constitute several useful technical and conceptual advances for our understanding of how bacterial cell division can be reconstituted from minimal components.

Despite these strengths, there are several rather significant weaknesses. These include a lack of proper explanations in some cases, and too much emphasis on data that largely confirm previous studies instead of breaking new ground. For example, Zieske and Schwillie (2014) already showed that oscillating MinCDE could focus FtsZ polymers into a fairly tight medial

band in cell-shaped lipid microcompartments, but oddly that report was not cited here. As a result of these confirmatory experiments, many of the figures seem quite repetitive.

Perhaps most importantly, the large continuous FtsA-attached FtsZ polymer bundles that completely encircle the liposomes here are very different from the tight complexes of treadmilling FtsZ-FtsA observed at the septum of dividing *E. coli* cells. Consequently, it is not clear whether the membrane deformation observed here is in any way relevant to forces on the membrane in walled bacterial cells such as *E. coli*.

Hopefully the comments below will be helpful to improve the impact of the paper.

We thank the reviewer for their comments and apologize for the apparently inappropriate discussion of prior work, which was caused by the wish to limit the discussion to sufficiently recent findings on these well-studied systems. We have outlined this in the general comments, and included some additional sentences in the manuscript. At the same time, we have added more explanations about the mechanical force exerted by FtsZ-ring in the manuscript, and this point is mainly discussed in the answer for the question 9), 15)-18), and no. 12 for reviewer #1. We are grateful for the constructive feedbacks on our own previous work and made minor corrections to the manuscript which we believe helped to improve the overall impact of this paper.

Major comments:

1) The authors do not adequately explain the different types of Min oscillations to the potential general readers of Nat Comm. In particular they need to explain what “pulsing” is.

We apologize for the inadequate explanation of Min wave dynamics inside lipid vesicles. We have included confocal images of each MinCDE wave mode in Supplementary Fig. 2a together with detailed information in the legend.

2) Some of Fig. 1 and the first part of the Results are confirmatory—e.g. the need for the MinD:E ratio to be 1:1, the static membrane localization if ratio is too high, or lumen localization if the ratio is too low. Although these confirmatory experiments show that the system is working as expected, they do not advance our understanding of the phenomena being studied.

Despite the other study that has studied a similar system (Kohyama et al., 2019, eLife), we believe our experimental setup differs sufficiently to provide new insights and advances to the current understanding of Min dynamics, assuring that the complete system (MinCDE proteins) works inside lipid vesicles (for the detailed discussion, please see the answer for reviewer #1 no. 2). More importantly, Fig. 1e shows that macromolecular crowding did not significantly affect the frequencies on the Min dynamics inside lipid vesicles, which has never been studied, but became critical when inducing FtsZ filaments under the crowding environments. However, we have reshaped the Fig 1 and moved the original Fig. 1c to the Supplementary Fig. 2c and d together with some additions to improve the overall readability.

3) Lines 229-231: this is not a major advance, as co-reconstitution of purified FtsZ-mts with MinCDE was done by Zieske and Schwille (2014).

We apologize that the previous version lacked the comparison or even citation with our previous study, where we have reconstituted the Min pole-to-pole oscillation together with FtsZ-mts condensation into the middle of PDMS fabricated chamber. This experimental setup is broadly used and offers major advantages when it comes to study membrane systems, however, it lacks some critical features of a cell model, which have a significant impact on the FtsAZ and MinCDE protein dynamics, first and foremost, the absence of a full membrane enclosure. For instance, the previous reports showed cell-sized confinement significantly alters the environmental requirements to induce Min waves (Kohyama et al. 2019, eLife), and also

membrane deformability plays an essential role to study FtsZ dynamics (Osawa and Erickson, 2013, Plos. Biol., Godino et al. 2019, Nat. Comm., Ganzinger et al., 2020, Angew. Chem. Int. Ed., Ramirez-Diaz et. al., 2021, Nat. comm.).

Therefore, considering the molecular dynamics of Min and Fts proteins, it could be expected that their cooperative behavior is also far away from what we could see in the previous studies. In this regard, our lipid vesicle-based reconstitution system is critically important to faithfully mimic the cellular environment such as cell-sized confinement and membrane flexibility. Obviously, this is the first time to our knowledge that reconstitutes the FtsZ-ring structure with MinCDE system inside such lipid vesicles as cell-like compartments. However, now we added the proper citations and comparisons between the previous studies (l. 272-279, 312-313, 337-338, 475-479, 495-499, and Supplementary discussion l. 240-252) to clarify that it is a major advance to reconstitute the FtsZ-mts/MinCDE system inside lipid vesicles.

4) Line 232: In vitro reconstitution with FtsA and FtsZ was done in ref. 23.

In this study, our major goal was to reconstitute the whole division ring placement system, but not the part of the system such as FtsA-FtsZ filaments. Hence, we believe it was reasonable to demonstrate the FtsA-FtsZ reconstitution in our experimental setup, which is certainly different from previous studies (Osawa and Erickson, 2013, PNAS, Godino et al. 2020, Comm. Biol.). Indeed, with our real-time observation, we succeeded to find new dynamics of FtsZ depending on the FtsA expression (please see the answer to the first major concern from reviewer #2). However, regarding other comments, we revised the manuscript and therefore the results of reconstituted FtsA-FtsZ system have been moved to the supplementary results (Supplementary Fig. 5b-e) to improve the readability.

5) Line 265-267: It seems that Ref. 31 already reconstituted the “cytoskeleton system” inside lipid vesicles via cell-free expression. If not, the authors need to explain how their advance is significant.

As we mentioned in answer 4) and to the first major concern from reviewer #2), we were able to visualize the real-time dynamics of the FtsZ filaments on the membrane and therefore consider it a major advance compared to previous studies. Now we further detailed on this point for clarity, as follows “Thus, we conclude that our experimental setup fully supports the transition to cell-free expression of our functional machineries, and that cell-free expressed FtsA enables us to capture the real-time dynamics of the development of an FtsZ-FtsA meshwork inside lipid vesicles.” (l. 297-300).

6) Line 284: the causality at this point is not fully backed by evidence, although later (line 297, supplementary Fig. 4d) there is a good correlation between pole to pole oscillations and proper FtsZ localization.

In both cases, there was a clear causality, although the assembled structure was not as stable as in the second case, which makes it a bit difficult to observe the phenomenology by eye. However, now we added Fig. 3d to show that the formation of FtsZ ring-like structure is strictly governed by pole-to-pole oscillation of the Min waves, which, as we hope, improves clarity. We thank the reviewer for this advice and hope that they will agree with our conclusions.

7) Line 288: are the traveling waves described here equivalent to the “circling” patterns made by Min proteins in liposomes described in Ref. 31?

We are sorry for the inconsistency of the technical term among the related studies. Yes, we would assume that they represent the same dynamics. We revised the manuscript to resolve the confusion (Supplementary file, l. 94-96, Supplementary Fig. 2a and legend of Supplementary Fig. 2).

8) Line 296-300: none of these findings/conclusions are surprising or new.

We agree with the reviewer that this statement is not surprising, although it represents the first confirmation of a ring-like structure formation process *in vitro* in cell-like models. As the reviewer may agree on, even the most compelling hypothesis is worth nothing without clear experimental evidence, which has strictly been partly lacking. A previous study (Zieske, Chwastek, and Schwille, 2016, *Angew. Chem. Int. Ed.*) described only the displacement of FtsZ filaments driven by Min waves inside lipid droplets, and therefore, our work provides a reproducible and quantitatively significant demonstration of this biologically relevant phenomenon inside cell-like models. Consequently, we included some sentences “Similar to the previous studies that indicate antagonistic membrane localization of FtsZ and MinCDE waves” (l. 312-313) and “Together with the previous studies showing the reorganization of FtsZ by Min waves, it became obvious that Min waves strictly govern FtsZ patterns, and more importantly, out of the two major dynamic Min patterns, only pole-to-pole oscillations, but not traveling waves, support stable FtsZ-ring formation inside lipid vesicles.” (l. 337-340) to clarify this point.

9) FtsA seems to be required for the membrane deformations in larger vesicles, as FtsZ-MTS on its own did not deform them. However, the dynamic treadmilling of FtsZ within the polymer bundles in the vesicles probably drive membrane deformations.

We completely agree with this concern, please see the answer no. 10 for the reviewer #1 for the explanation of membrane-binding protein aided deformation of vesicles. Nevertheless, as the reviewer mentioned, we believe that FtsZ treadmilling itself is also able to deform lipid membranes under the appropriate conditions. Indeed, it has been stated that FtsZ-Venus-mts can also deform membranes on deflated vesicles (Ramirez-Diaz et al., 2021, *Nat. comm.*) (please also see the next comment), which explains our lack of FtsZ deformations when using isosmotic conditions. On the other hand, in our cell-free expression system, we observed a higher degree of vesicle deformation in larger vesicles (Fig. 4f), which seems to be directly related to a higher concentration of FtsZ (Supplementary Fig. 6h and i). Therefore, we could assume that a higher concentration of FtsZ might be able to exert greater constriction forces and naturally, it becomes attractive to study concentration-dependent deformation in a controllable manner especially using FtsZ-mts.

10) Do the authors think that the FtsZ swirls observed when bound to FtsA in SLBs are also occurring on liposome membranes? Given the larger scale of the polymer bundles compared with the swirls, this seems unlikely, but then how relevant are the swirls or the straight polymer bundles to what happens *in vivo*?

We would like to thank the reviewer for the comment and interesting observation. It is known that Mg^{2+} ions significantly strengthen the self-association and assembly of FtsZ both in solution and membrane (Monterroso et al., 2013, *Methods*, Rivas et al., 2013, *Biophys. Rev.*). For instance, our previous study (Ramirez-Diaz, 2018, *Plos. Biol.*) found that FtsZ-mts on SLBs forms clear polymer bundles at 5 mM of Mg^{2+} , while forming swirling vortices at 1 mM of Mg^{2+} , showing that Mg^{2+} controls the FtsZ-mts dynamics on membrane. In this regard, PURE reaction buffer typically contains 13 mM $Mg(OAc)_2$, and such high concentration of Mg concentration may affect the FtsZ dynamics to invoke strong lateral interaction, which leads the assembly of polymer bundles rather than swirling. However, since Mg is critical for the transcription-translational enzymatic activities (Fujiwara and Doi, 2016, *PLoS One*), it is technically challenging to express proteins under the low Mg concentration buffers. Hence, it is not feasible to test this hypothesis in our experimental setup, although it is quite intriguing to further characterize the FtsZ dynamics and therefore would be a key goal for in future biophysical experiments. The *in vivo* relevance of the FtsZ dynamics and the possible role of bundles/swirls inside the cell are intriguing and still quite underexplored.

The surface density of FtsZ on the membrane is also an important factor for the assembly of swirls or bundles. As the FtsZ concentration on the membrane is variable among the cell cycle, it might also determine the dynamics and the structures that FtsZ could assemble (Mannik et al., 2018, Mol. Microbiol.). As observed *in vitro*, FtsZ swirls are only formed at low protein density while bundles are favored at high, which could reassemble what happens *in vivo* (Loose and Mitchison, 2014, Nat. Cell Biol., Ramirez-Diaz, 2018, Plos. Biol.). Then, studies of different FtsZ structures on the membrane might be interesting to understand the molecular mechanism of the system *in vivo*, and the differential dynamics of swirls/bundles are interesting to understand the whole process. Despite all the efforts and studies involving FtsZ *in vivo*, the role of FtsZ in the constriction of the cell membrane is still under debate. It seems likely that FtsZ-rings and swirls can exert forces by the treadmilling dynamics of the filaments, but it is still challenging to study such processes either *in vivo* or *in vitro*.

11) Line 426 and following: do the authors have an explanation for why pole to pole oscillations transition into traveling waves and then static localization? Does MinE specifically lose function or get degraded over time? Does the ATP in the system get exhausted?

In this case, the pattern transition was likely invoked by the increasing concentration of MinDE proteins over time, rather than ATP consumption or protein dysfunctionalization. First, PUREflex contains ATP/GTP regeneration system to prolong the cell-free expression up to 6h in general usage, assuring that ATP is maintained at sufficient concentration to induce Min waves within this time scale as discussed in the manuscript (up to 90 min). Second, we indeed captured the same trend of wave patterns also in the relatively simpler MinCDE expression system in Fig. 3c and Supplementary Fig. 6b, suggesting that a change on MinDE concentration determines the wave pattern. In addition, time-dependent transition of wave patterns was also caused by the concentration shift of Min proteins by PURE cell-free expression observed in related studies (Yoshida et al., 2019, Chem. Sci., Takada et al., 2022, Sci. Adv.), and also suggested in a theoretical study (please see the next comment).

12) It is not clear how Min oscillations initiate in perfectly spherical liposomes with an aspect ratio of 1.0, as there is no defined long axis until there is some deformation as shown by the authors, and compartment geometry has been shown previously to determine the orientation and organization of Min oscillations. How do back and forth oscillations get started without some asymmetry?

It is known that even when the Min proteins are encapsulated in spherical vesicles, pole-to-pole oscillation can emerge, presumably due to tiny environmental fluctuations inducing spatiotemporal symmetry breaking. Indeed, in our results (Fig. 1d, e and Supplementary Fig. 2) and previous *in vitro* studies (Litschel et al., 2018, Angew. Chem. Int. Ed., Godino et al., 2019, Nat. comm.), we could find pole-to-pole oscillations in spherical vesicles. Moreover, it was described that pole-to-pole oscillations can emerge inside spherical containers in a theoretical study by Kohyama et al. 2019, eLife, and more recently, Takada et al., 2022, Sci. Adv. In the latter case, it was explained that MinE concentration and the MinD's ATPase stimulation activity of MinE are determinant conditions for the assembly of wave patterns inside closed environments. Since the authors succeeded to predominantly induce pole-to-pole oscillations inside water-in-oil droplets in this report, it is plausible to believe that the Min system intrinsically has a tendency to tip towards pole-to-pole oscillation.

13) Line 493-494: What is an "expected insight"? MinCDE pole-to-pole oscillations are already known to be directly responsible for FtsZ ring formation and placement *in vivo* (and by Zieske and Schwille 2014, among others).

We are sorry for the confusing sentences here, but we actually "expected" this Min wave driven FtsZ-ring structure that can be observed both *in vivo* and *in vitro*. we have added a sentence and references to clarify the point in accordance (l. 475-476).

14) Line 501-503: Does the positive feedback on Min oscillation caused by the condensed FtsZ band depend on MinC? The prediction would be that it would be MinC-dependent given the direct interaction between MinC and FtsZ.

Please see the answer to the reviewer #1, no. 4, 5, and 6.

15) As discussed on lines 533-534, the membrane deformation with FtsZ-FtsA inside liposomes reported here is basically the same as the membrane deformation with FtsZ-FtsA* rings in liposomes reported in ref. 23, but just more frequent and efficient. Even the tubular liposomes reported 14 years ago in ref. 17 with FtsZ-MTS-Venus exhibited focused FtsZ “rings” that sometimes deformed the membranes to create a constriction.

As the reviewer highlighted, it is not new that FtsZ is able to exert forces and constrict membranes by forming small swirls or even big ring-like structures (Osawa et al., 2008, *Science*, Osawa and Erickson, 2013, *PNAS*, Godino et al. 2020, *Comm. Biol.*, Ramirez-Diaz et. al., 2021, *Nat. comm.*). However, our results are not only more clearly reproducible and controlled, but we were also able to reconstitute the whole process of FtsZ-ring placement, including the assembly and positioning driven by Min waves, capturing the real-time dynamics. None of these features were achieved in previous studies and therefore, they constitute major advantages of our study.

16) Line 548: The decreased surface-to-volume ratio in larger spheres is an attractive hypothesis to explain the greater tendency of larger liposomes to undergo FtsZ-mediated constriction. However, it remains puzzling how extensive FtsZ bundles on the liposomes of very large diameters 10x the size of an *E. coli* cell could exert constriction forces, yes extensive FtsZ bundles of liposomes that are only modestly smaller (but still much larger than bacteria) would not.

Please see the answer for the reviewer 1, no. 9.

17) Lines 559-560: “spatiotemporal imbalance of membrane curvature” needs to be explained better.

We apologize for the confusing sentence. It has been described before (Litschel et al., 2018, *Angew. Chem. Int. Ed.*, Christ et al., 2021, *Soft Matter*) that MinDE binding to the membrane inside vesicles exerts a mechanical effect over the membrane, causing a visible dynamic deformation of vesicles. In this case, the periodic relocation of the Min proteins in the lumen and on the membrane can undergo mechanical changes of the membrane and even demonstrating fission and fusion-like dynamic deformation of vesicles. In this regard, we rationalized that membrane binding of Min proteins only from the inner leaflet of lipid vesicles, but not from outside might induce sufficient deformation of lipid vesicles. To clarify this point, we revised the manuscript to “Additionally, our previous study showed that reversible membrane attachment of MinDE proteins to the inner leaflet of the lipid vesicles through their amphipathic helices induces a spatiotemporal imbalance of membrane curvature” (l.531-533).

18) Line 562-564: The potential of ZipA to enhance membrane deformations by FtsZ/FtsA is not explained in sufficient detail, particularly as *E. coli* can still divide quite normally in the absence of ZipA when suppressor mutant divisome proteins are made.

In this case, we referred to ZipA as a candidate to improve the deformations by FtsZ as a topic of interest for membrane biophysics rather than implying a biological meaning of the molecular system. ZipA contains a transmembrane region instead of a binding membrane domain as FtsA or MinD (Moy et al. 2000, *Biochemistry*). For the case of ZipA or other transmembrane proteins, their attachment might be able to bend the membrane as hypothesized previously (Derganc & Čopič, 2016, *Biochim. Biophys. Acta – Biomembr.*). Thus, attachment of FtsZ through ZipA to the membrane might provide an extra degree of freedom to improve

deformations by FtsZ torsion and treadmilling. Future studies might address the combination of FtsA and ZipA to potentially improve the system leading to a higher degree of membrane deformation. However, we stated that we referred to use ZipA as a bending inducer by a biophysical aspect (l. 535-540).

19) The supplemental movies are very interesting to watch, and it is impressive how many liposomes have successful FtsZ medial localization. However, in the final two movies, it is puzzling why the largest and most prominent liposomes, initially still, start getting jostled around when the Min oscillations start (or perhaps vice versa). Can the authors explain this sudden onset of jerky liposome motions that coincide with the beginning of Min oscillations (and the focusing of FtsZ polymers towards the midpoint)?

We would like to thank the reviewer for their positive assessment of our system. In this case, for the supplemental movies 7 and 8, we indeed observed not only the membrane constriction by FtsZ-ring, but also the back and forth movement of the lipid vesicle following the Min oscillation. As we mentioned above in answer 18), membrane attachment-detachment of Min proteins can induce further membrane curvature from inside by exerting a mechanical force. Hence, together with the pole-to-pole oscillation, Min wave might provoke the periodical back and forth movements of lipid vesicles.

Minor comments:

- 1) L. 243 should be “device to”
- 2) L. 256 should be “...process of FtsZ bundle...”
- 3) L. 261 should be “...the vesicle was gradually decreased...”
- 4) L. 262 should be “...while an increase of...”
- 5) L. 263-264: delete the first instance of “in larger vesicles”
- 6) L. 272 and 275: replace “emerged” with “resulted in”
- 7) L. 278: please cite a reference for “as expected”
- 8) L. 283: delete “a”
- 9) L. 292: should be “therein”
- 10) L. 417-418: should be ...”which maintained the Min oscillations in a pronounced...”
- 11) L. 437: delete “on”
- 12) L. 517: should be “identifying”
- 13) L. 573: replace “notorious” with “daunting”
- 14) L. 728 should be “chamber”
- 15) L. 747 should be “the vesicle periphery”

We thank for that detailed correction on our manuscript. We accepted all suggestions from the reviewer in the main/supplementary manuscript.

REVIEWERS' COMMENTS

Reviewer #1 (Remarks to the Author):

The authors have made a major effort to write a detailed response letter explaining why in their view the findings are novel and why they provide new insight, addressing each comment of the reviewers, sometimes at length. They have also added some valuable experiments to the manuscript concerning the feedback mechanism which strengthens their interpretation regarding the interplay between the FtsZ and MinDCE systems, and they have corrected sub-optimal citations and clarified the experimental conditions in the figure legends which is appreciated.

Surprisingly however, they did not really address much the major conceptual criticism of the reviewers in their main text. Introduction and Discussion are mostly unchanged, which indicates that the manuscript may indeed be intended for the specialist in the field. Alternatively, the decision of the authors to leave their main text mostly unchanged may indicated that the manuscript may be better suited for a journal allowing a longer Introduction and Discussion, in case space constraints prevented the authors to be more clear in their manuscript about novelty, conceptual advance, and mechanistic insight.

Reviewer #2 (Remarks to the Author):

NA

Reviewer #3 (Remarks to the Author):

I am happy with the authors' responses to my critiques and the other reviewers, and the revisions have made the paper easier to read and more focused on the novel results. The MinC mutant data also nicely support their conclusions from other data. I do think that the Supplemental Discussion is too long and overlaps with the main Discussion, but I will not object if the authors wish to keep it as is.

I have a few minor suggestions below to improve the writing.

Main text:

Line 426: replace "take its origin" with "originate"

Line 534: replace "Mins" with "Min proteins"

Line 537: replace “In this study it is” with “In that study it was”

Line 539: replace “module” with “model”

Line 556: I suggest replacing the existing phrase with “the nucleoid, which could exert strong effects on the membrane through its replication and segregation”

Line 647: replace “In case” with “For”

Supplemental:

Lines 91, 531: should be “fluorescent”

Line 92: replace “placed in” with “fused to”

Line 174: replace “under a fluorescent microscopy” to using fluorescence microscopy”

Line 183: Explain that this is a “sandwich” fusion with the FP inserted between 55 and 56.

Line 183: should be “As this chimera was reported as...”

Line 242: should be “fully confined”

Line 725: Authors’ names have typos.

Finally, was there a specific reason why mScarlet-MinC was used for some experiments and mCherry-MinC for others?

Reviewer #1 (Remarks to the Author):

The authors have made a major effort to write a detailed response letter explaining why in their view the findings are novel and why they provide new insight, addressing each comment of the reviewers, sometimes at length. They have also added some valuable experiments to the manuscript concerning the feedback mechanism which strengthens their interpretation regarding the interplay between the FtsZ and MinDCE systems, and they have corrected sub-optimal citations and clarified the experimental conditions in the figure legends which is appreciated.

Surprisingly however, they did not really address much the major conceptual criticism of the reviewers in their main text. Introduction and Discussion are mostly unchanged, which indicates that the manuscript may indeed be intended for the specialist in the field. Alternatively, the decision of the authors to leave their main text mostly unchanged may indicate that the manuscript may be better suited for a journal allowing a longer Introduction and Discussion, in case space constraints prevented the authors to be more clear in their manuscript about novelty, conceptual advance, and mechanistic insight.

We thank the reviewer for their acknowledgment of our major efforts to rationalize our approach and main scientific results in our rebuttal letter. Indeed, we had significantly edited the discussion and results sections to address the reviewer's comments, and included new important results regarding positive feedback between Min wave and FtsZ-ring formation. Also, we had moved a substantial portion of results and original discussion to the supplement and included some introductory sentences regarding the positive feedback between MinCDE and FtsZ systems.

While we agree that the introduction and discussion parts, even after our careful revision, may still have remained too bulky, the conclusion that the reviewer draws from it, i.e. that the manuscript was better suited for another journal, seems rather oblique – after all, it is the relevance of the results and not the writing that motivate submission to certain journals. Nevertheless, by a further significant reduction of dispensable information in Introduction and Discussion, thereby highlighting the main findings even more explicitly, we are now convinced that our manuscript will be of great interest to a much larger audience than just the specialists in the field. The topic of bottom-up synthetic biology based on GUVs, and also the Min/FtsZ system are without any doubt sufficiently well-known to appeal to a large readership of Nature Communications. But also beyond this community, the great advance represented by our work with regard to earlier attempts to build a minimal divisome should be sufficiently evident.

Reviewer #2 (Remarks to the Author):

NA

Reviewer #3 (Remarks to the Author):

I am happy with the authors' responses to my critiques and the other reviewers, and the revisions have made the paper easier to read and more focused on the novel results. The MinC mutant data also nicely support their conclusions from other data. I do think that the Supplemental Discussion is too long and overlaps with the main Discussion, but I will not object if the authors wish to keep it as is.

We would like to thank the reviewer for their positive assessment and appreciation of our efforts made to improve the manuscript. We are aware that some of the results shown in the supplementary notes might overlap with the main discussion in some aspects, although extra explanation in certain points might be useful to provide additional information for readers in checking the supplementary material in detail.

I have a few minor suggestions below to improve the writing.

Main text:

Line 426: replace “take its origin” with “originate”

Line 534: replace “Mins” with “Min proteins”

Line 537: replace “In this study it is” with “In that study it was”

Line 539: replace “module” with “model”

Line 556: I suggest replacing the existing phrase with “the nucleoid, which could exert strong effects on the membrane through its replication and segregation”

Line 647: replace “In case” with “For”

Supplemental:

Lines 91, 531: should be “fluorescent”

Line 92: replace “placed in” with “fused to”

Line 174: replace “under a fluorescent microscopy” to using fluorescence microscopy”

Line 183: Explain that this is a “sandwich” fusion with the FP inserted between 55 and 56.

Line 183: should be “As this chimera was reported as...”

Line 242: should be “fully confined”

Line 725: Authors’ names have typos.

We thank the reviewer for detailed corrections and have accepted all suggestions in both main and supplementary manuscript, although some of them have been rephrased, deleted or moved to the supplementary discussion by further edits.

Finally, was there a specific reason why mScarlet-MinC was used for some experiments and mCherry-MinC for others?

In principle, mScarlet-I is more than three times brighter red fluorescent protein than mCherry (brightness 15.84 for mCherry vs. 56.16 for mScarlet-I) and therefore is more suitable for use as a purified protein. At the same time, we did not observe any major difference in the MinCDE dynamics by using either mCherry- or mScarlet-I-MinC in the purified system. However, in cell-free expression systems, folding of nascent amino acid chain would be also a critical feature to obtain properly matured and functional proteins in vitro environment. In this regard, mCherry has a better folding property than mScarlet-I (15 min for mCherry vs. 36 min for mScarlet-I) and indeed, we tested both mCherry-MinC and mScarlet-I-MinC proteins in cell-free system and found slightly better brightness in mCherry-MinC as a result. Hence, we chose different fluorescent proteins for visualization of MinC among different experimental setups for obtaining the best results.